**SPECIAL ISSUE**
**CELL BIOLOGY OF THE NUCLEUS**

# Paclitaxel compromises nuclear integrity in interphase through SUN2-mediated cytoskeletal coupling

Thomas Hale, Victoria L. Hale, Piotr Kolata, Ália dos Santos* and Matteo Allegretti*

## ABSTRACT

Regulation of lamin A/C levels and distribution is crucial for nuclear integrity and mechanotransduction via the linker of nucleoskeleton and cytoskeleton (LINC) complex. Dysregulation of lamin A/C correlates with poor cancer prognosis, and its levels determine sensitivity to the microtubule-stabilising drug paclitaxel. Paclitaxel is well-known for disrupting mitosis, yet it also reduces tumour size in slow-dividing tumours, indicating an additional, poorly characterised interphase mechanism. Here, we reveal that paclitaxel induces nuclear aberrations in interphase through SUN2-dependent lamin A/C disruption. Using advanced optical imaging and electron cryo-tomography, we show the formation of aberrant microtubule–vimentin bundles during paclitaxel treatment, which coincides with nuclear deformation and altered lamin A/C protein levels and organisation at the nuclear envelope. SUN2 is required for lamin A/C reduction upon paclitaxel treatment and is in turn regulated by polyubiquitylation. Furthermore, lamin A/C expression levels determine not only cell survival during treatment but also recovery after drug removal. Our findings support a model in which paclitaxel acts through both defective mitosis and interphase nuclear–cytoskeletal disruption, providing additional mechanistic insights into a widely used anticancer drug.

KEY WORDS: Cancer, Nucleus, LINC, SUN2, Lamin A/C

## INTRODUCTION

Paclitaxel is a taxane widely used to treat breast, ovarian and lung cancers (Bishop et al., 1999; Ozols et al., 2003; Ranson et al., 2000). It acts by binding stoichiometrically to β-tubulin within microtubules and inducing conformational changes that stabilise interactions between adjacent α-tubulin–β-tubulin heterodimers (Xiao et al., 2006), preventing microtubule depolymerisation and enhancing bundle formation (Schiff and Horwitz, 1980; Turner and Margolis, 1984). Historically, the anti-cancer mechanism of paclitaxel was thought to rely on its suppression of microtubule dynamics in the mitotic spindle, which activates the mitotic checkpoint to result in cell cycle arrest and subsequently apoptosis (Jordan et al., 1993). Since then, it has been shown that

cells can escape mitotic arrest in clinically relevant concentrations of paclitaxel, resulting in multipolar divisions due to improper chromosome segregation, ultimately leading to cell death (Weaver, 2014; Zasadil et al., 2014). However, paclitaxel reduces tumour size even in tumours with low duplication rates and where only a small fraction of cells are proliferative (Milross et al., 1996; Orth et al., 2011; Janssen et al., 2013). Furthermore, intravital imaging of tumours in mice treated with taxanes shows that cell death is predominantly induced independently of mitotic defects (Janssen et al., 2013). This suggests that the anti-cancer mechanism of paclitaxel likely involves an additional activity in interphase (Komlodi-Pasztor et al., 2011; Fürst, 2013; Smith et al., 2021).

There have been several hypotheses proposing how paclitaxel might induce apoptosis in interphase, including the activation of pro-apoptotic signalling following the disruption of microtubule structure (Wang et al., 1999; Huang et al., 2000), and perturbation to mitochondria (André et al., 2000; Lu et al., 2011). As microtubules perform myriad functions in the cell, it is likely that perturbations to multiple different aspects of cellular function contribute to apoptosis. Recently, Smith et al. (2021) have shown that paclitaxel treatment leads to fragmentation of the nucleus independently of cell division (Smith et al., 2021, 2022; Smith and Xu, 2021). Together with observations that paclitaxel results in microtubule reorganisation around the nucleus during interphase (Banerjee et al., 1997; Rosenblum and Shivers, 2000) and ectopic localisation of nuclear envelope (NE) and lamina proteins in cancer cells (Theodoropoulos et al., 1999), this suggests that an additional anti-cancer mechanism of paclitaxel might be via the disruption of nuclear–cytoskeletal coupling, leading to an increased risk of DNA damage and consequent cell death (Denais et al., 2016; Takaki et al., 2017; Cho et al., 2019). However, this mechanism is poorly understood.

Nuclear–cytoskeletal coupling is mediated by linker of nucleoskeleton and cytoskeleton (LINC) complexes, which consist of a trimer of SUN proteins that span the inner nuclear membrane and interact at the C-terminal end of a periplasmic coiled-coil region with the C-terminus of three KASH protein monomers (Starr and Fridolfsson, 2010). KASH proteins traverse the outer nuclear membrane and connect to the cytoskeleton in the cytoplasm at their N-terminus (Mellad et al., 2011), whereas the N-terminus of SUN proteins connects to the nuclear lamina in the nucleoplasm (Haque et al., 2006). Cytoskeletal forces are therefore transmitted to the nuclear lamina and linked chromatin domains via these direct connections, which provide the physical basis for mechanotransduction (Bouzid et al., 2019).

The nuclear lamina plays an important role in controlling the mechanical response of the nucleus, ensuring it can withstand and react to mechanical forces (Dahl et al., 2004; Lammerding et al., 2006). The nuclear lamina consists of the A-type lamins lamin A and C (lamin A/C), which are splice isoforms encoded by the *LMNA* gene, as well as the B-type lamins lamin B1 and B2 (Burke and Stewart, 2013). These

Structural Studies Division, MRC Laboratory of Molecular Biology, Cambridge CB2 0QH, UK.

*Authors for correspondence (adossantos@mrc-lmb.cam.ac.uk; matteoall@mrc-lmb.cam.ac.uk)

T.H., 0009-0001-7155-6351; V.L.H., 0000-0002-7769-7251; Á.d.S., 0000-0002-3176-8317; M.A., 0000-0002-9542-7482

are intermediate filament proteins that polymerise to form a filamentous mesh beneath the NE (Burke and Stewart, 2013). Aberrations to the nuclear lamina and nuclear morphology are prevalent across many cancer types and are often used as biomarkers in diagnosis (Abel et al., 2024). In particular, lamin A/C expression is frequently altered in cancer cells (Wu et al., 2009; Belt et al., 2011; Gong et al., 2015; Matsumoto et al., 2015; Bell and Lammerding, 2016), and this is thought to play a role in migration and invasion, a hallmark of cancer that leads to metastasis (Bell and Lammerding, 2016). However, alterations to the nuclear lamina might also make cancer cells particularly sensitive to perturbations to nuclear–cytoskeletal coupling. In support of this, decreased lamin A/C expression has been shown to increase sensitivity to paclitaxel in ovarian cancer cells (Smith et al., 2021).

Here, we propose that cancer cells have increased vulnerability to paclitaxel both during interphase and following aberrant mitosis due to pre-existing defects in their NE and nuclear lamina. Supporting this, we show that paclitaxel disrupts the organisation of microtubule, actin and vimentin filaments around the nucleus during interphase leading to nuclear deformation. Nuclear deformation was more severe in cells with decreased levels of lamin A/C. We also report similar nuclear deformation phenotypes in cells where microtubules were stabilised by overexpression of Tau, thus linking the observed nuclear deformation upon paclitaxel treatment with its microtubule-bundling activity. Our data show that paclitaxel treatment severely affects nuclear lamina organisation and that this occurs via connection to SUN2-containing LINC complexes, which were regulated by ubiquitylation following paclitaxel treatment. Finally, we show that lamin A/C expression levels affect the overall sensitivity to paclitaxel and, importantly, recovery from it.

Altogether, our data provides insights into interphase mechanisms of action for paclitaxel and the role of the nuclear lamina in drug sensitivity.

## RESULTS
### Paclitaxel induces cytoskeletal reorganisation around the nucleus during interphase
To investigate how paclitaxel affects the organisation of microtubules during interphase, human fibroblasts were treated with 5 nM paclitaxel or control medium for 16 h. Experiments were performed at 5 nM paclitaxel (with additional experiments to determine dose relationships at 1 and 10 nM) because this aligns with previous studies (Jordan et al., 1993; Theodoropoulos et al., 1999; Smith et al., 2021). Furthermore, previous analysis of plasma from individuals being treated with paclitaxel reveals that typical concentrations are within the low nanomolar range (Zasadil et al., 2014), and concentrations of 5–10 nM are required in cell culture to reach the same intracellular concentrations observed *in vivo* in patient tumours (Weaver, 2014). This aligns with *in vitro* cytotoxic studies of paclitaxel in eight human tumour cell lines which show that paclitaxel's IC50 ranges between 2.5 and 7.5 nM (Liebmann et al., 1993).

Following paclitaxel treatment, cells were analysed by immunofluorescence confocal microscopy for α-tubulin and stained for DNA with Hoechst 33342 (Fig. 1A). In contrast to what was seen in control cells, where microtubules were dispersed throughout the cytoplasm and emanated from a single microtubule-organising centre (MTOC), in paclitaxel-treated cells, the microtubules formed dense rings around the nucleus with no single obvious MTOC (Fig. 1A). To determine whether these microtubules surrounding the nucleus were bundled, we used two-dimensional (2D) stochastic optical reconstruction microscopy (STORM) for high-resolution

morphological characterisation (Fig. 1B). Quantitative HDBSCAN clustering of STORM data revealed an increase of 44% in the microtubule bundle diameter in the presence of paclitaxel compared to that seen in untreated cells (Fig. 1C).

Given that there is extensive crosstalk between microtubule, actin and intermediate filaments (Oberhofer et al., 2020; Pradeau-Phélut and Etienne-Manneville, 2024), we next tested how the organisation of actin and vimentin was affected by paclitaxel treatment (Fig. 1D,E). As expected, actin filaments were distributed throughout the cortex in control cells, but in the presence of paclitaxel, perinuclear actin was instead condensed into puncta (Fig. 1D). Interestingly, vimentin colocalised strongly with microtubules (Gurland and Gundersen, 1995; Prahlad et al., 1998), and also reorganised into a dense network surrounding the nucleus following paclitaxel treatment (Fig. 1E). The extent of the cytoskeletal reorganisation correlated with the concentration of paclitaxel, as microtubules became increasingly bundled and the actin cytoskeleton more condensed with increasing concentrations (0 to 10 nM) of paclitaxel (Fig. S1A). Similar cytoskeletal reorganisation around the nucleus was also observed in human breast cancer MDA-MB-231 cells in 5 nM paclitaxel (Fig. S1B).

To visualise the rearranged cytoskeleton at high resolution and investigate how it interacts with the nucleus, we used electron cryo-tomography (cryo-ET). Cells were first thinned by cryo-focused ion beam (cryo-FIB) milling before tilt-series acquisition on the resulting lamellae. In the reconstructed tomograms, paclitaxel-treated cells frequently contained large, dense bundles of parallel microtubules and vimentin filaments (Fig. 1F; Fig. S1C), which were absent from control cells (Fig. S1D). These bundles could be observed to closely associate with the NE (Fig. 1F; Fig. S1C).

Overall, these data indicate that paclitaxel induces large-scale reorganisation of the cytoskeleton, beyond just microtubule filaments, in the perinuclear region that could lead to changes in nuclear-cytoskeletal coupling following paclitaxel treatment.

### Paclitaxel-induced nuclear deformation is dependent on lamin A/C expression levels
LINC complexes connect the cytoskeleton to the NE and underlying lamina (Starr and Fridolfsson, 2010). We hypothesised that the observed cytoskeletal reorganisation would affect this mechanical coupling and force distribution to the nucleus, resulting in nuclear deformation. To test this, high-content live-cell imaging was performed over 48 h in the presence of 0, 1, 5 or 10 nM paclitaxel to measure nuclear solidity of nuclei in wild-type fibroblasts (Fig. 2A,B; Fig. S2A,B). Nuclear solidity provided a measure of nuclear deformation (Janssen et al., 2022) (Fig. S2A) on a scale of 0 (most deformed) to 1 (least deformed), with Hoechst 33342 used as a nuclear marker. As expected, nuclear solidity decreased over time in paclitaxel-treated cells, and this nuclear deformation was concentration dependent (Fig. 2A,B). Decreased nuclear solidity upon paclitaxel treatment was also observed in slow dividing, serum-starved cells, thus suggesting that the observed nuclear deformation occurs independently of cell division (Fig. 2C; Fig. S2C,D).

As A-type lamins are major determinants of nuclear rigidity and the ability of the nucleus to withstand mechanical forces (Broers et al., 2004), we next tested how lamin A/C expression levels affect paclitaxel-induced nuclear deformation. Nuclear solidity was compared between wild-type cells, cells transiently overexpressing GFP–lamin A and cells with lamin A/C depleted using siRNA (Fig. 2D,E; Fig. S2E). In the absence of paclitaxel, nuclear solidity remained high across all three lamin A/C expression levels. However,

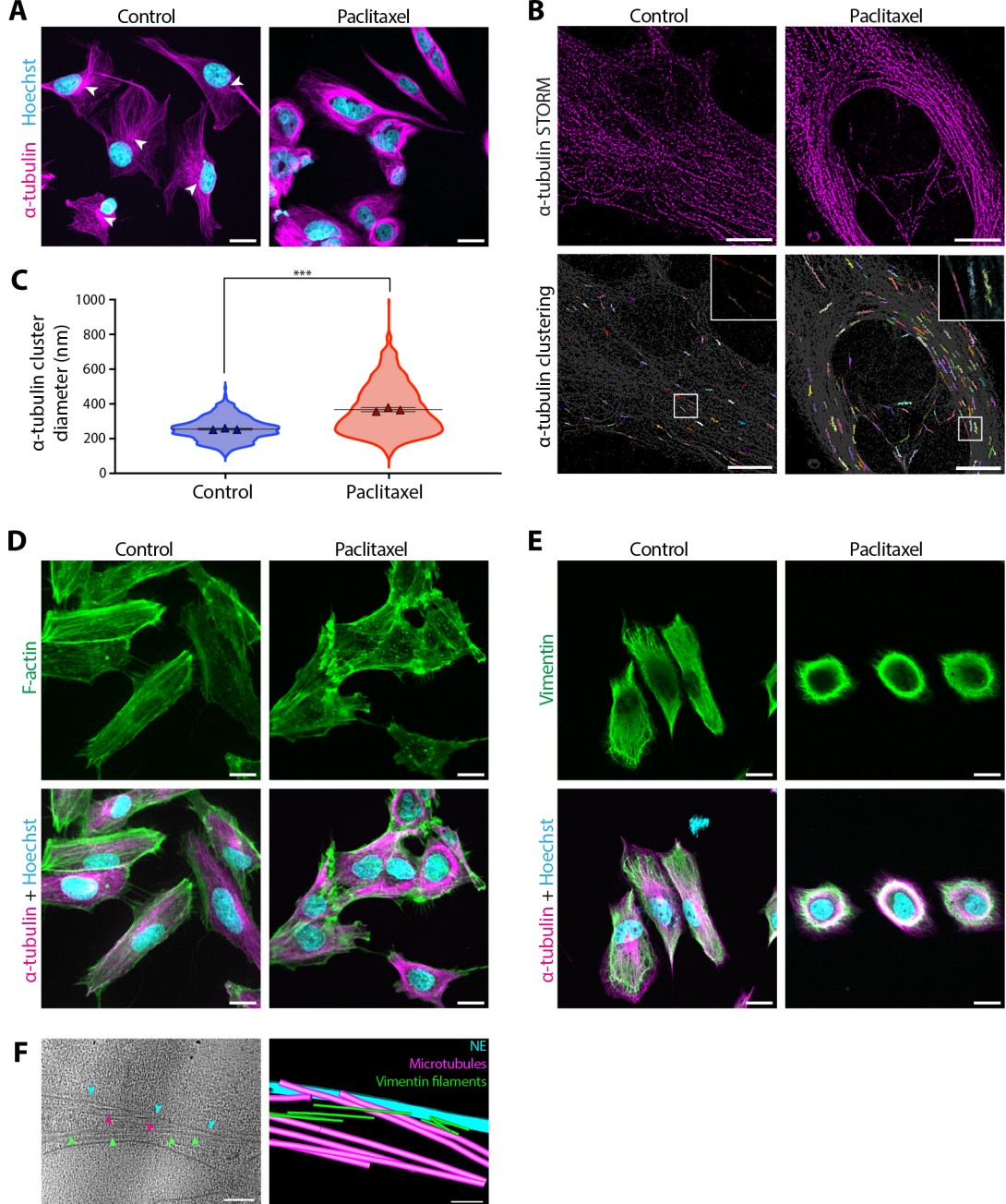

**Fig. 1. Paclitaxel-induced cytoskeletal reorganisation around the nucleus in interphase.** (A) Confocal micrographs of cells fixed after 16 h incubation in control medium or 5 nM paclitaxel. DNA was stained using Hoechst 33342 (cyan), and microtubules using α-tubulin immunofluorescence (magenta). MTOCs are marked with arrowheads. Scale bars: 20 μm. (B) STORM imaging of α-tubulin immunofluorescence in cells fixed after 16 h incubation in control medium or 5 nM paclitaxel. Lower panels show α-tubulin clusters generated with HDBSCAN analysis. Different colours distinguish individual α-tubulin clusters, representing individual microtubule filaments or filament bundles. Scale bars: 10 μm. (C) Violin plot comparing the diameter of α-tubulin clusters from B between control (blue) and paclitaxel-treated cells (red). The mean is shown in black with error bars showing the s.e.m. from three biological repeats ($n$=3). The mean of each biological repeat is marked with a triangle and consists of >300 measurements from five or more cells. ***$P$=0.00012 (unpaired two-tailed $t$-test). (D,E) Confocal micrographs of control and paclitaxel-treated cells stained for DNA using Hoechst 33342 (cyan), microtubules using α-tubulin immunofluorescence (magenta), and either (D) F-actin using Alexa Fluor 488–phalloidin (green) or (E) vimentin using immunofluorescence (green). Scale bars: 20 μm. (F) Left, 2D slice of a reconstructed tomogram of the NE region in a paclitaxel-treated cell. Bundled vimentin filaments (green arrowheads) and microtubules (magenta arrowheads) are seen closely apposed to the NE (cyan arrowheads). Right, segmentation of the NE (cyan), microtubules (magenta) and vimentin filaments (green) from the tomogram. Scale bars: 100 nm. Images in A and D–F are representative of three biological repeats each with >50 cells.

in the presence of paclitaxel, nuclear deformation was significantly greater in lamin A/C-depleted cells and significantly reduced when lamin A was overexpressed (Fig. 2E). Overall, this indicates that paclitaxel-induced nuclear deformation depends on lamin A/C expression levels.

In addition to misshapen nuclei, cells containing multiple smaller nuclei, termed multimicronuclei, were observed following paclitaxel treatment (Fig. S2F). Live-cell imaging showed that in every instance they arose immediately following cell division, but we never observed this during interphase (Fig. S2G). Importantly, quantifying the

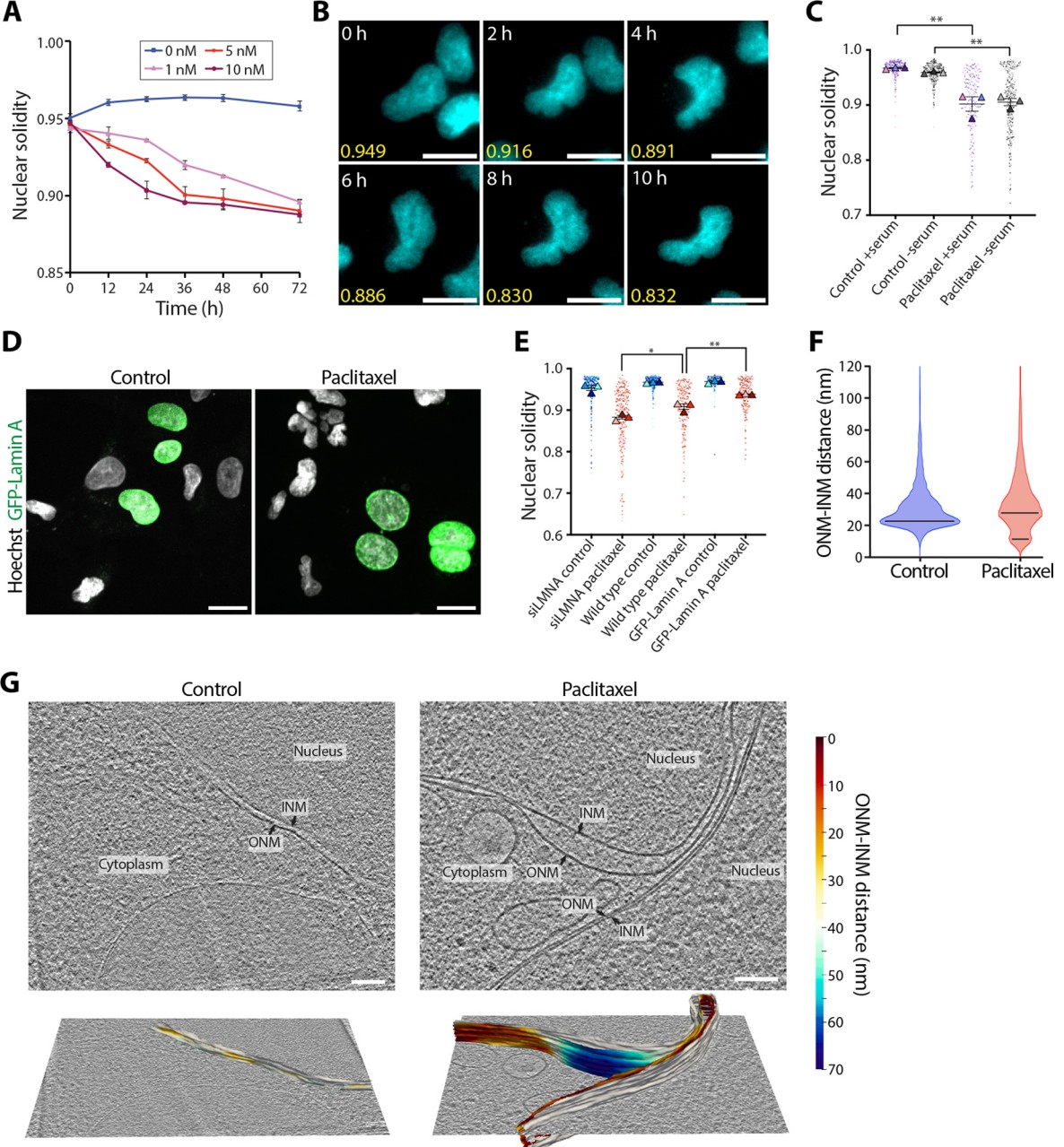

**Fig. 2. Paclitaxel treatment results in nuclear deformation.** (A) Nuclear solidity of Hoechst 33342-stained nuclei over 72 h in 0, 1, 5 or 10 nM paclitaxel. Error bars show the s.e.m. from three biological repeats ($n$=3), each with more than 50 cells. (B) Example frames from a live-cell time-series used for the nuclear solidity analysis in A, showing Hoechst 33342-stained nuclei (cyan) and respective nuclear solidity measurements (yellow) taken every 2 h at 0–10 h after the addition of 5 nM paclitaxel. Scale bars: 20 µm. (C) Quantification of nuclear solidity from cells cultured with complete medium (+serum) or serum-starved medium (−serum) in control conditions or after 16 h in 5 nM paclitaxel. The mean is shown in black with error bars showing the s.e.m. from three biological repeats ($n$=3), each with at least 50 cells. The datapoint from each cell is marked with a dot colour-coded according to the biological replicate it came from, with the mean of each biological repeat marked with a triangle of the same colour. \*\*$P$=0.0075 (−serum); \*\*$P$=0.0011 (+serum) (unpaired two-tailed $t$-test for control versus paclitaxel). (D) Confocal micrographs of cells transiently transfected with GFP–LMNA to overexpress GFP–lamin A (green), fixed after 16 h incubation in control medium or 5 nM paclitaxel. Cells were stained for DNA using Hoechst 33342 (grey). Scale bars: 20 µm. (E) Quantification of nuclear solidity from wild-type cells, cells with lamin A/C knocked down (siLMNA), or cells with GFP–lamin A overexpressed, following 16 h incubation in control medium or 5 nM paclitaxel. For GFP–lamin A overexpression samples, only cells expressing GFP-lamin A were analysed. Error bars show s.e.m. from three biological repeats ($n$=3), each with at least 30 cells. The datapoint from each cell is marked with a dot colour-coded according to the biological replicate it came from, with the mean of each biological repeat marked with a triangle of the same colour. \*$P$=0.0311 (wild-type paclitaxel versus siLMNA paclitaxel), \*\*$P$=0.0099 (wild-type paclitaxel versus GFP–lamin A paclitaxel) (unpaired two-tailed $t$-test). (F) Violin plot comparing the ONM–INM distance between control and paclitaxel-treated cells, quantified from nuclear membrane segmentations from high-resolution tomograms using surface morphometrics (Barad et al., 2023). A total NE area of 1.64 µm² consisting of 1,236,316 datapoints over 23 tomograms from 6 cells was analysed for the control, and 3.97 µm² consisting of 3,781,706 datapoints over 41 tomograms from 9 cells was analysed for paclitaxel-treated cells. The black lines represent the modal values (21.0 nm for control; 10.5 nm and 25.5 nm for paclitaxel). (G) Example 2D slices of reconstructed tomograms of the NE in control and paclitaxel-treated cells. Segmentations of the ONM and INM that were used for the morphometrics analysis in F are shown in the lower panels, with the ONM coloured using a heatmap of the ONM–INM distance. Scale bars: 100 nm.

proportion of cells undergoing mitosis following 30 h incubation in 5 nM paclitaxel using immunofluorescence analysis showed that the proportion of mitotic cells upon paclitaxel treatment was 6.9±0.4% (mean±s.e.m.), only slightly higher than that in control conditions (4.4±0.6%) (Fig. S2H). This suggests that at clinically relevant concentrations, paclitaxel does not induce sustained mitotic arrest, and cells are instead able to exit mitosis. However, immunofluorescence microscopy analysis of paclitaxel-treated cells revealed that this mitosis was defective, as dividing cells contained disorganised mitotic spindles that were frequently multipolar, in contrast to bipolar spindles in untreated cells (Fig. S2I). Overall, our data indicates that the nuclear deformation seen upon paclitaxel treatment occurs both during interphase and following defective mitosis.

The appearance of multimicronuclei in ovarian cancer cells was previously observed to depend on decreased lamin A/C expression levels (Smith et al., 2021). In line with this, we found that the frequency of multimicronucleated cells after 24 h in 5 nM paclitaxel was significantly lower when lamin A was overexpressed and significantly higher when lamin A/C was knocked down compared to the wild type (Fig. S2J).

To investigate changes to nuclear ultrastructure caused by paclitaxel, we used cryo-ET. Tomographic data of the NE showed altered intermembrane spacing between the inner and outer nuclear membranes (INM and ONM, respectively) following paclitaxel treatment (Fig. 2F,G; Fig. S2K). In non-treated cells, this distance was uniformly ∼25 nm (Kucińska et al., 2023), but in paclitaxel-treated cells we frequently observed NE areas where the membranes appeared to have a large separation or by contrast, were in very close proximity. To quantify this, we used morphometrics analysis (Barad et al., 2023) of the NE segmented from tomograms of control and paclitaxel-treated cells (Fig. 2F,G; Fig. S2K). In control cells, 67% of the total NE area analysed had a spacing of 20–40 nm, whereas in paclitaxel-treated cells this was decreased to 49% (Fig. 2F). This was due to the significantly greater variability in the ONM–INM distance in paclitaxel-treated cells (Levene test; $P<0.0001$) because of the increased frequency of loci where the membranes were in very close proximity or highly separated, as seen in Fig. 2G and Fig. S2K. Indeed, the ONM–INM distance showed a broader and bimodal distribution in paclitaxel-treated cells, with a peak within the expected range at 25.5 nm and an additional peak at 10.5 nm where the two membranes were in very close proximity (Fig. 2F). This contrasted with the narrow distribution around the single peak at 21.0 nm for the control (Fig. 2F). Together, this indicates that perturbation to the cytoskeleton due to paclitaxel treatment results in deformation to the overall nuclear shape and to the NE, resulting in altered NE spacing.

### Paclitaxel treatment disrupts the nuclear lamina via the LINC complex protein SUN2

Next, we investigated whether the nuclear lamina is also impacted by paclitaxel treatment. Immunofluorescence analysis of lamin A/C and lamin B1 showed an irregular and patchy nuclear distribution of both proteins in paclitaxel-treated cells, compared to control cells (Fig. 3A; Fig. S3A). Interestingly, western blotting revealed that total cellular levels of lamin A/C were decreased by paclitaxel in a concentration-dependent manner (Fig. 3B,C) but, in contrast, lamin B1 levels were not significantly affected (Fig. 3B,C).

Reverse transcription quantitative PCR (RT-qPCR) showed that the paclitaxel-induced decrease in lamin A/C levels did not occur at the mRNA level (Fig. S3B), indicating that regulation occurs at the protein level instead. Lamin A/C can be phosphorylated during interphase at several sites (Kochin et al., 2014), and this can alter its assembly into the lamina network and overall stability (Bertacchini

et al., 2013). However, immunoblotting of whole-cell lysates against phospho-Ser404 lamin A/C, a major site of lamin A/C phosphorylation during interphase (Kochin et al., 2014), showed no changes in the phosphorylation levels after paclitaxel treatment (Fig. S3C,D). Phos-tag gel analysis corroborated these findings, revealing no changes in the overall phosphorylation state of lamin A/C (Fig. S3E,F).

We also tested the acetylation and ubiquitylation states of lamin A/C, as these are known to affect its protein stability and assembly (Borroni et al., 2018; Karoutas et al., 2019). However, immunoprecipitation experiments against lamin A/C followed by western blotting for acetyl-lysine and polyubiquitin showed no significant change in the acetylation or ubiquitylation state of lamin A/C in paclitaxel-treated cells (Fig. S3G,H). This suggests an alternative mechanism for the observed reduction in lamin A/C levels following paclitaxel treatment.

We hypothesised that perturbation to lamin A/C upon paclitaxel treatment could instead occur through its direct connection to the cytoskeleton mediated via LINC complexes, in particular via SUN proteins (Haque et al., 2006; Starr and Fridolfsson, 2010). To test this, we used confocal microscopy to analyse the colocalisation of SUN1 and SUN2 with lamin A/C (Fig. 3D,E; Fig. S3I,J). Our data show that, following paclitaxel treatment, both SUN1 and SUN2 became unevenly distributed at the NE, resulting in a patchy appearance (Fig. 3D; Fig. S3I). However, although SUN2 retained high colocalisation with lamin A/C, SUN1 showed decreased colocalisation in paclitaxel-treated cells (Fig. 3D,E; Fig. S3I,J). Furthermore, western blotting revealed that SUN2 protein levels were decreased in a concentration-dependent manner (Fig. 3F,G), whereas SUN1 showed no significant reduction in total protein levels (Fig. 3F,G). As the effects of paclitaxel on SUN2 parallel those of lamin A/C, this suggests that there is a close interplay between these two proteins upon paclitaxel treatment.

Western blot analysis of serum-starved cells showed a similar decrease in lamin A/C and SUN2 levels following paclitaxel treatment (Fig. 4A–C), indicating that this perturbation is not strictly dependent on cell division. Therefore, we hypothesised that the paclitaxel-induced changes to lamin A/C protein levels occur through its direct connection to the cytoskeleton via SUN2-containing LINC complexes. Indeed, in contrast to the decrease of lamin A/C levels in paclitaxel-treated wild-type cells, lamin A/C levels remained high across all paclitaxel concentrations tested in SUN2-knockdown cells (Fig. 4D,E; Fig. S4A). In the absence of SUN2, transmission of forces from the cytoskeleton to the nuclear lamina could be impaired and therefore be less able to perturb lamin A/C levels. However, when lamin A/C is knocked down, a concentration-dependent decrease in SUN2 protein levels is still observed upon paclitaxel treatment (Fig. S4B,C). Altogether, our results indicate that SUN2-containing LINC complexes are essential for sensing and transmitting cytoskeletal forces to the nuclear lamina upon paclitaxel treatment.

RT-qPCR showed that the decrease in SUN2 levels following paclitaxel treatment did not occur at the mRNA level (Fig. S4D). Mechanical regulation of SUN2 is mediated by ubiquitylation by the E3 ubiquitin ligase SCF$^{\beta TrCP}$ (Krshnan et al., 2022). To test whether the paclitaxel-induced decrease in SUN2 levels is mediated by SUN2 ubiquitylation, we pulled down SUN2 from whole-cell lysates of control or paclitaxel-treated cells and performed western blotting for polyubiquitin C (Fig. 4F,G). This showed that there was a significant increase in SUN2 ubiquitylation following paclitaxel treatment (Fig. 4F,G).

Finally, we confirmed that these results apply to cancer cells. As expected, paclitaxel treatment also resulted in a significant decrease

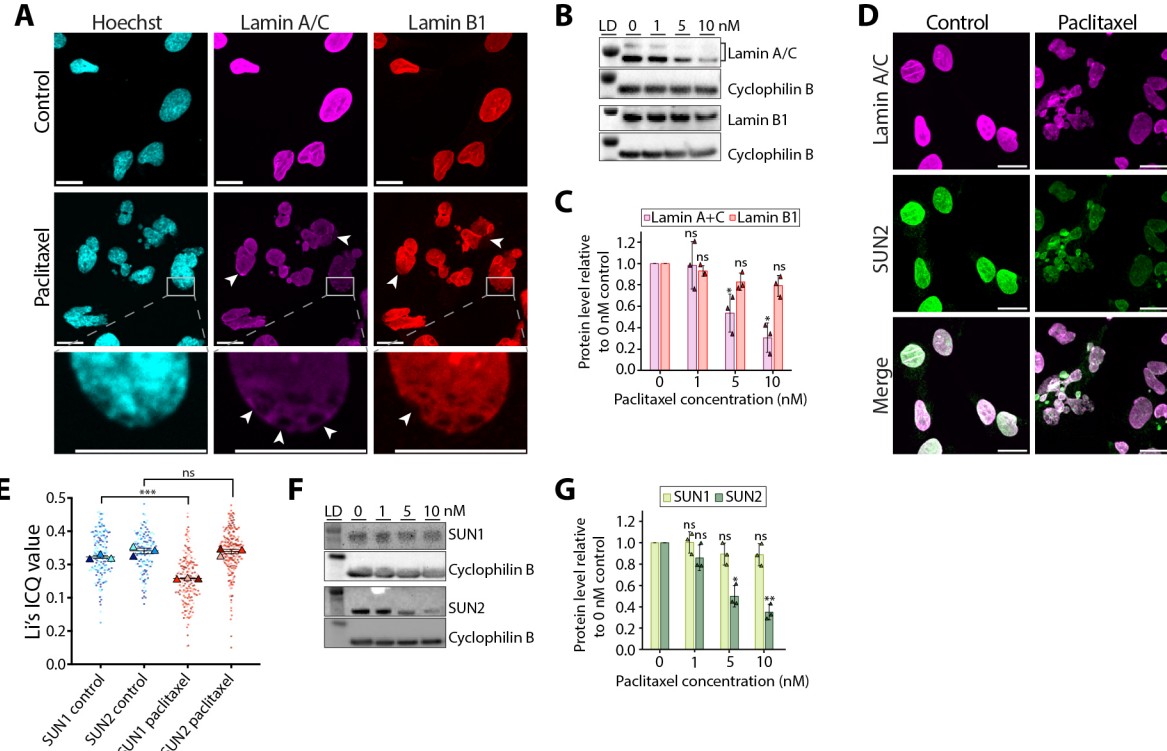

**Fig. 3. Paclitaxel treatment results in aberrant organisation and decreased levels of lamin A/C and SUN2.** (A) Confocal micrographs of cells fixed after 16 h incubation in control medium or 5 nM paclitaxel. DNA was stained using Hoechst 33342 (cyan), and lamin A/C (magenta) and lamin B1 (red) were stained using immunofluorescence. Areas of the lamina that are patchy or have lamin A/C or B1 missing are marked with arrowheads. Lower panels show magnified views of a patchy area of the lamina. Scale bars: 20 μm. (B) Western blots for lamin A/C and lamin B1 of whole-cell lysates following 16 h incubation in medium containing 0, 1, 5 or 10 nM paclitaxel. Cyclophilin B was used as a loading control. Lane 1 shows protein ladder (LD; 75 kDa for lamin A/C and lamin B1 panels, and 25 kDa for cyclophilin B panels). (C) Quantification of lamin A/C and lamin B1 protein levels from B. Each band was normalised to the corresponding cyclophilin B loading control. Error bars represent the s.d. from three biological repeats ($n=3$) which are marked with triangles. ns, not significant; *$P<0.05$ (lamin A/C, 1 nM $P=0.9059$, 5 nM $P=0.0452$, 10 nM $P=0.0128$; lamin B1, 1 nM $P=0.139$, 5 nM $P=0.0665$, 10 nM $P=0.0656$) (one-sample $t$-test with null hypothesis mean=1). (D) Confocal micrographs of cells fixed after 16 h incubation in control medium or 5 nM paclitaxel. Cells were stained for lamin A/C (magenta) and SUN2 (green) using immunofluorescence. Scale bars: 20 μm. (E) Analysis of the colocalisation between lamin A/C and SUN1 or SUN2. Fluorescence colocalisation was quantified using Li's ICQ value (Li et al., 2004), where more positive values represent better positive colocalisation. Error bars represent the s.e.m. from three biological repeats ($n=3$), each with more than 30 cells. The datapoint from each cell is marked with a dot colour-coded according to the biological replicate it came from, with the mean of each biological repeat marked with a triangle of the same colour. ns, not significant (SUN2, $P=0.8788$); ***$P=0.0002$ (SUN1) (unpaired two-tailed $t$-test control versus paclitaxel). (F) Western blots for SUN1 and SUN2 of whole-cell lysates following 16 h incubation in medium containing 0, 1, 5 or 10 nM paclitaxel. Cyclophilin B was used as a loading control. Lane 1 shows protein ladder (LD; 98 kDa for SUN1 and SUN2 panels, and 28 kDa for cyclophilin B panels). (G) Quantification of SUN1 and SUN2 protein levels from F. Each band was normalised to the corresponding cyclophilin B loading control. Error bars represent the s.d. from three biological repeats ($n=3$) which are marked with triangles. ns, not significant; *$P<0.05$; **$P<0.01$ (SUN1, 1 nM $P=0.9607$, 5 nM $P=0.2142$, 10 nM $P=0.1988$; SUN2, 1 nM $P=0.1794$, 5 nM $P=0.0133$, 10 nM $P=0.0042$) (one-sample $t$-test with null hypothesis mean=1). Images in A are representative of three biological repeats, each with >50 cells.

in nuclear solidity (Fig. S4E), a patchy nuclear lamina (Fig. S4F), and decreased lamin A/C and SUN2 levels in breast cancer MDA-MB-231 cells (Fig. S4G–I).

## Nuclear deformation and a patchy nuclear lamina are specific to microtubule stabilisation

To test whether the nuclear deformation and nuclear lamina disruption observed in interphase cells after paclitaxel treatment occur specifically due to microtubule bundling, we overexpressed Tau, a neuronal microtubule-associated protein known to induce microtubule stabilisation and bundling (Monroy-Ramírez et al., 2013). Cells transiently overexpressing GFP–Tau showed the presence of large microtubule bundles, similar to what was seen in paclitaxel-treated cells (Fig. 5A). However, unlike paclitaxel treatment, which induced microtubule bundling into rings around the nucleus (Fig. 1A), the microtubule bundles in cells overexpressing GFP–Tau were distributed throughout the cytoplasm (Fig. 5A). Interestingly, in cells where Tau-induced microtubule bundles were in

close proximity to the nucleus, we observed substantial nuclear shape deformation (Fig. 5A) and a significant decrease in nuclear solidity (Fig. 5B). Furthermore, GFP–Tau overexpression also resulted in an uneven distribution of both lamin A/C and lamin B1 (Fig. 5C), confirming that perturbation to the nuclear lamina during interphase is caused by microtubule stabilisation.

## Lamin A/C expression levels determine sensitivity to paclitaxel treatment

As lamin A/C expression levels determine nuclear deformation upon paclitaxel treatment, we then tested how this relates to overall cellular sensitivity to paclitaxel. We performed high-content screening to quantify cell confluency of the wild-type cells, and cells with lamin A/C knocked down or GFP–lamin A overexpressed over 48 h in 0, 1, 5 or 10 nM paclitaxel (Fig. 6A). Without paclitaxel, cell growth was similar across all three lamin A/C expression levels tested. In 1 and 5 nM paclitaxel, cell growth was reduced, but to a significantly lesser extent when lamin A was overexpressed compared to the wild-type

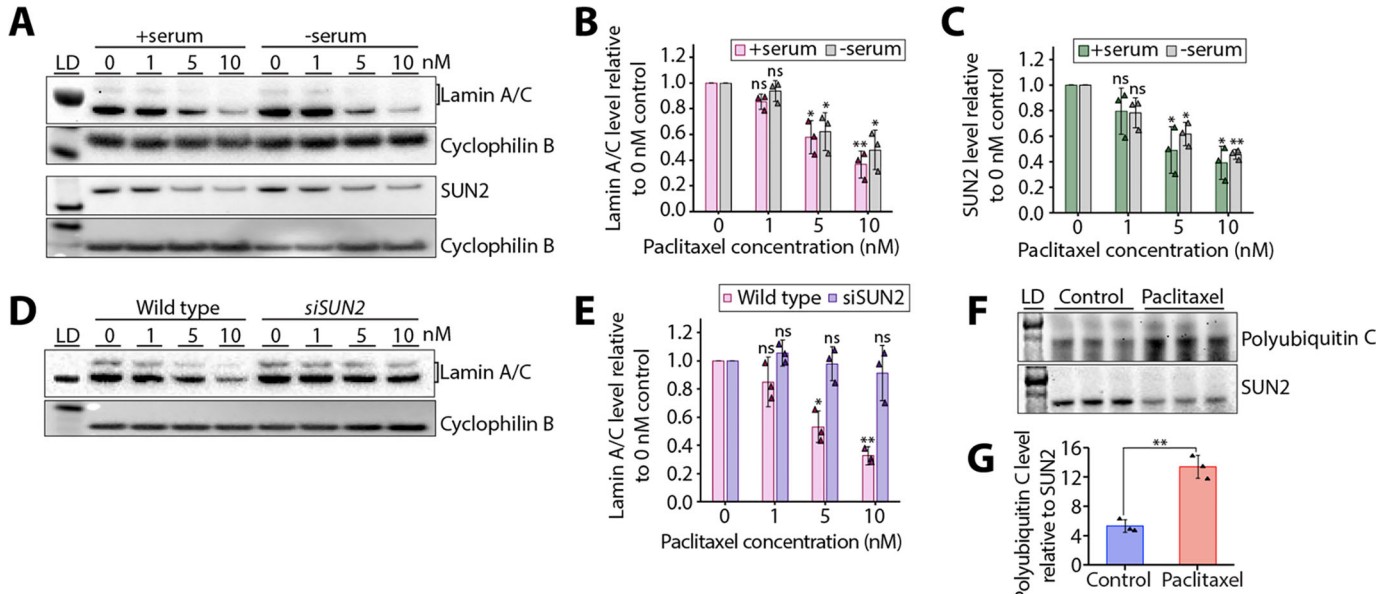

**Fig. 4. The paclitaxel-induced decrease in lamin A/C levels occurs via SUN2.** (A) Western blots for lamin A/C and SUN2 of whole cell lysates from cells cultured in complete medium (+serum) or serum-starved medium (−serum) and following 16 h incubation in 0, 1, 5 or 10 nM paclitaxel. Cyclophilin B was used as a loading control. Lane 1 shows protein ladder (LD; 75 kDa for lamin A/C panel, 20 kDa for cyclophilin B upper panel, 98 kDa for SUN2 panel and 28 kDa for cyclophilin B lower panel). (B,C) Quantification of lamin A/C (B) and SUN2 (C) protein levels from A. Each band was normalised to the corresponding cyclophilin B loading control. Error bars represent the s.d. from three biological repeats ($n$=3) which are marked with triangles. ns, not significant; *$P$<0.05; **$P$<0.01 (lamin A/C +Serum, 1 nM $P$=0.0504, 5 nM $P$=0.0301, 10 nM $P$=0.0092; lamin A/C −serum: 1 nM $P$=0.3119, 5 nM $P$=0.0472, 10 nM $P$=0.0272; SUN2 +serum, 1 nM $P$=0.1904, 5 nM $P$=0.0404, 10 nM $P$=0.0146; SUN2 −serum, 1 nM $P$=0.0849, 5 nM $P$=0.0182, 10 nM $P$=0.0015) (one-sample $t$-test with null hypothesis mean=1). (D) Western blots for lamin A/C of whole-cell lysates from wild-type or SUN2-knockdown (siSUN2) cells following 16 h incubation in 0, 1, 5 or 10 nM paclitaxel. Cyclophilin B was used as a loading control. Lane 1 shows protein ladder (LD; 62 kDa for lamin A/C panel, and 28 kDa for cyclophilin B panel). (E) Quantification of lamin A/C protein levels from D. Each band was normalised to the corresponding cyclophilin B loading control. Error bars represent the s.d. from three biological repeats ($n$=3) which are marked with triangles. ns, not significant; *$P$<0.05; **$P$<0.01 (wild type, 1 nM $P$=0.2779, 5 nM $P$=0.0185, 10 nM $P$=0.0032; siSUN2, 1 nM $P$=0.3984, 5 nM $P$=0.8031, 10 nM $P$=0.5198) (one-sample $t$-test with null hypothesis mean=1). (F) Top panel, western blot for polyubiquitin C following pull-down of SUN2 from whole cell lysates of control cells or cells treated with 5 nM paclitaxel for 16 h. Three biological repeats were used for each condition ($n$=3). Bottom panel, to control for SUN2 protein levels, the same membrane was stripped and blotted for SUN2. (G) Quantification of polyubiquitin C levels from F, with each band normalised to SUN2. Error bars represent the standard deviation from three biological repeats ($n$=3) which are marked with triangles. **$P$=0.0014 (unpaired two-tailed $t$-test control versus paclitaxel).

and lamin A/C knockdown (Fig. 6A). In 10 nM paclitaxel, cell confluency remained low across all three lamin A/C expression levels (Fig. 6A), as the high concentration prevented almost any cell growth.

Next, we tested the effects of lamin A/C expression levels on cell viability in paclitaxel (Fig. 6B) by calculating the proportion of dead or apoptotic cells using the caspase-3 and -7 dye FLICA (Fig. S5A). Increasing concentrations of paclitaxel resulted in lower cell viability, and this was more pronounced in lamin A/C-depleted cells (Fig. 6B). However, at high concentrations of paclitaxel (10 nM), lamin A/C knockdown no longer had an observable impact on cell viability. Conversely, cells overexpressing lamin A displayed higher cell viability in paclitaxel, even at high concentrations (Fig. 6B). Overall, these data indicate that lamin A/C expression levels affect sensitivity to paclitaxel in terms of both cell growth and cell death.

Finally, we tested recovery from paclitaxel treatment. Immunofluorescence confocal microscopy showed that microtubule bundling was reverted following paclitaxel removal (Fig. S5B). Importantly, the localisation of lamin B1 and lamin A/C (Fig. S5B) and the protein levels of lamin A/C and SUN2 also recovered after removal of 5 nM paclitaxel (Fig. S5C,D).

To assess the effect of lamin A/C levels on paclitaxel recovery, we also measured cell confluency of wild-type and lamin A/C-knockdown cells following removal of paclitaxel (Fig. 6C). Whereas wild-type cells expanded in cell number following paclitaxel removal, lamin A/C-knockdown cells showed little cell growth (Fig. 6C). Altogether, this shows that lamin A/C expression

levels also affect recovery from paclitaxel, in addition to cell death and inhibition of cell growth.

## DISCUSSION

Despite its widespread use, the mechanism by which paclitaxel selectively kills cancer cells is not yet fully understood. A better understanding of the effects of paclitaxel is therefore important for improving treatment and reducing drug resistance and toxicity. In this study, we investigated how paclitaxel disrupts nuclear–cytoskeletal coupling during interphase. We show that paclitaxel treatment results in nuclear aberrations, including NE deformation, changes to nuclear intermembrane spacing, multimicronucleation and disruption to the nuclear lamina via SUN2-containing LINC complexes. Cells with lower lamin A/C expression levels, typical of many different cancer cells (Wu et al., 2009; Belt et al., 2011; Gong et al., 2015; Matsumoto et al., 2015; Bell and Lammerding, 2016), showed increased sensitivity to paclitaxel in terms of cell growth, death and recovery, indicating how disrupted nuclear–cytoskeletal coupling can lead to a selective targeting of cancer cells, even in slowly proliferating tumours. Furthermore, we directly link nuclear aberrations to microtubule bundling around the nucleus.

The anti-cancer activity of paclitaxel was originally thought to be solely due to sustained mitotic arrest caused by disruption of the mitotic spindle (Jordan et al., 1993). Evidence for this came from *in vitro* cell culture and xenograft models, which rely on cells with extensive replicative capacity and short doubling times (Fuchs and

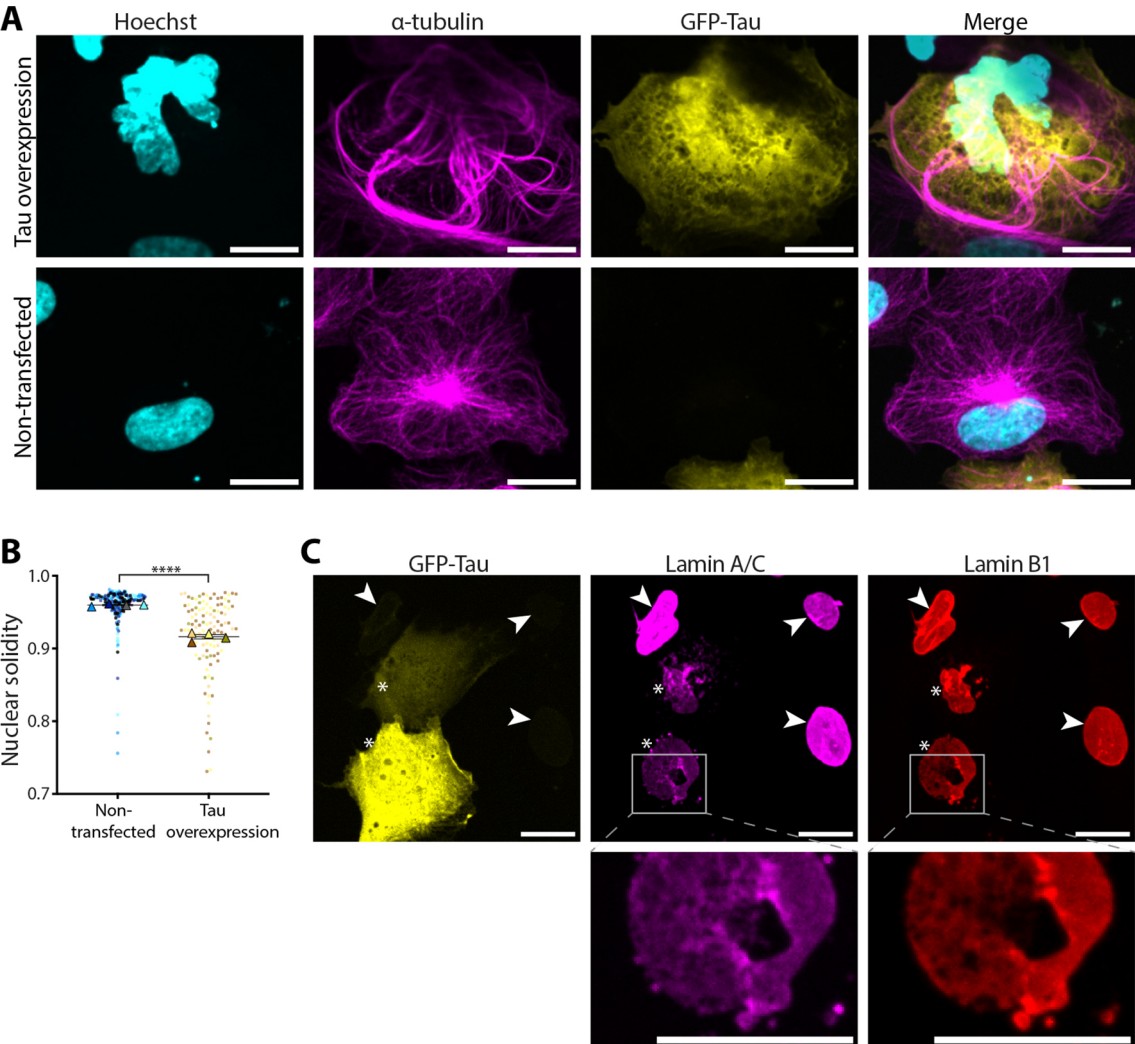

**Fig. 5. Microtubule stabilisation by GFP–Tau overexpression results in a patchy nuclear lamina and decreased nuclear solidity**. (A) Confocal micrographs of cells fixed 48 h after transient transfection with GFP–Tau (yellow). DNA was stained using Hoechst 33342 (cyan), and microtubules using α-tubulin immunofluorescence (magenta). The top panels show a cell overexpressing GFP–Tau; the bottom panels show a non-transfected control cell. Scale bars: 20 μm. (B) Bar chart comparing the nuclear solidity of non-transfected cells and cells overexpressing GFP–Tau. Error bars show the s.e.m. from four biological repeats (n=4), each with at least 60 cells. The datapoint from each cell is marked with a dot colour-coded according to the biological replicate it came from, with the mean of each biological repeat marked with a triangle of the same colour. ****$P$=7.68×10$^{-6}$ (unpaired two-tailed $t$-test). (C) Confocal micrographs of cells fixed 48 h after transient transfection with GFP–Tau (yellow). Lamin A/C (magenta) and lamin B1 (red) were stained using immunofluorescence. Cells overexpressing GFP–Tau are marked with asterisks, whereas non-transfected control cells are marked with arrowheads. Lower panels show magnified views of a patchy area of the lamina. Scale bars: 20 μm. Images in A and C are representative of three biological repeats, each with >30 cells.

Johnson, 1978; Jordan et al., 1993; Milas et al., 1995; Yamori et al., 1997). However, these cells might be more vulnerable to mitotic arrest than tumour cells in individuals with cancer, which can have longer doubling times (Komlodi-Pasztor et al., 2011; Kay et al., 2019). Furthermore, tumour biopsies from individuals with breast cancer treated with microtubule stabilising drugs show that mitotic cells are rare, despite patient response to therapy (Denduluri et al., 2007; Komlodi-Pasztor et al., 2011). One alternative hypothesis to explain this lack of mitotic cells in paclitaxel-treated tumour samples is that at intratumoral concentrations, cancer cells undergoing mitosis can escape the mitotic checkpoint, and this mitotic slippage results in defective chromosome segregation and apoptosis of the resulting aneuploid cells (Weaver, 2014; Zasadil et al., 2014). Interestingly, we show here that mitotic slippage in clinically relevant concentrations of paclitaxel can also occur in healthy human cells as the mitotic index of healthy human

fibroblasts remained low in paclitaxel and they could exit mitosis, resulting in multimicronucleation (Fig. S2F–H). However, the frequency of multimicronucleation, which always occurred following exit from mitosis (Fig. S2G), depended on lamin A/C expression levels (Fig. S2J), as previously reported in ovarian cancer cells (Smith et al., 2021). This is likely explained by the important role the nuclear lamina plays in the fragmentation and reassembly of nuclear membranes during mitosis (Gerace and Burke, 1988; Lopez-Soler et al., 2001). NE reformation around aberrantly segregated chromosomes could therefore be affected by lamin A/C expression levels, explaining the differences in multimicronucleation frequency and how cancer cells could be more susceptible due to their aberrant NEs that frequently show lower lamin A/C expression levels (Bell and Lammerding, 2016).

However, multimicronucleation alone cannot account for the activity of paclitaxel against tumours with slow duplication rates

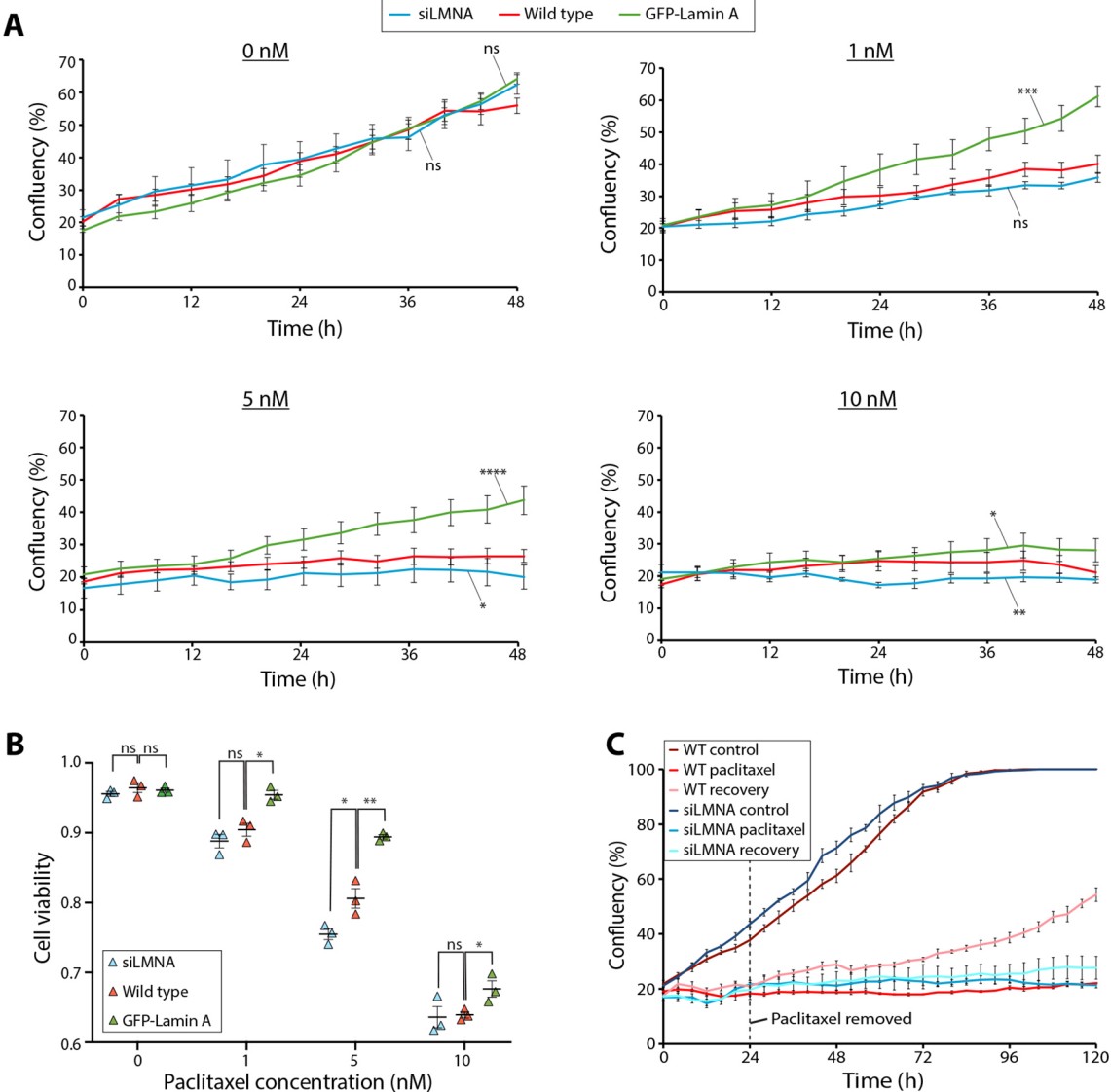

**Fig. 6. Lamin A/C expression level affects cell sensitivity to and recovery from paclitaxel.** (A) Cell confluency of wild-type (red), GFP–lamin A overexpression (green) and lamin A/C-knockdown (blue) cells over 48 h in 0, 1, 5 or 10 nM paclitaxel. Confluency was calculated from brightfield high-content live-cell images. Error bars represent the s.e.m. from five biological repeats. ns, not significant; *P<0.05; **P<0.01; ***P<0.001; **** P<0.0001 (0 nM, GFP–lamin A versus wild type P=0.3186, siLMNA versus wild type P=0.6074; 1 nM, GFP–lamin A versus wild type P=0.0003, siLMNA versus wild type P=0.1662; 5 nM, GFP–lamin A versus wild type P=4.5×10⁻⁵, siLMNA versus wild type P=0.0267; 10 nM, GFP–lamin A versus wild type P=0.0207, siLMNA versus wild type P=0.0029) (one-way three-factor repeated measures ANOVA with Bonferroni post hoc test). (B) Quantification of cell viability of wild-type (red), GFP–lamin A overexpression (green) and lamin A/C-knockdown (blue) cells following 48 h incubation in 0, 1, 5 or 10 nM paclitaxel. Cell viability was calculated by subtracting the proportion of the total number of cells that were dead/apoptotic (identified by FLICA staining) from 1. Error bars represent s.e.m. from three biological repeats (n=3) which are marked with triangles, each with at least 95 cells. ns, not significant; *P<0.05; **P<0.01 (0 nM, siLMNA P=0.3244, GFP–lamin A P=0.6742; 1 nM, siLMNA P=0.2911, GFP–lamin A P=0.0107; 5 nM, siLMNA P=0.0325, GFP-lamin A P=0.0036; 10 nM, siLMNA P=0.8297, GFP-lamin A P=0.0428) (unpaired two-tailed t-test versus wild type). (C) Cell confluency of wild-type and lamin A/C knockdown cells in control medium (control), medium with the sustained presence of 5 nM paclitaxel (paclitaxel) or medium from which paclitaxel was removed after 24 h (recovery). Error bars represent the s.e.m. from three biological repeats.

(Milross et al., 1996; Orth et al., 2011; Janssen et al., 2013), as it is unlikely that enough cells undergo mitosis in paclitaxel for this mechanism to be sufficient. This is supported by intravital imaging of murine tumours treated with docetaxel, another taxane closely related to paclitaxel, which demonstrated that apoptosis is induced in most tumour cells independently of aberrant mitosis, including both mitotic arrest and multimicronucleation (Janssen et al., 2013). For its activity against these cancers, paclitaxel must therefore have additional anti-cancer effects outside of mitosis (Komlodi-Pasztor et al., 2011; Fürst, 2013; Smith et al., 2021). In support of this, drugs

that specifically block mitosis have not been successful in the clinic, indicating that the efficacy of microtubule-targeting drugs like paclitaxel comes from additional effects in interphase as well as mitosis (Komlodi-Pasztor et al., 2011). Recent studies showing that paclitaxel affects the nuclear integrity of cancer cells in interphase indicate that this activity could be due to disruption of nuclear–cytoskeletal coupling (Smith et al., 2021, 2022; Smith and Xu, 2021).

Previously, paclitaxel has been shown to affect microtubule organisation in interphase cells in a cell-type-specific manner, with

some cells containing microtubule rings around the nucleus whereas others contain multiple microtubule asters (Banerjee et al., 1997). Here, we show that paclitaxel promotes the formation of dense bundles of microtubules and vimentin filaments surrounding the interphase nucleus in human fibroblasts (Fig. 1; Fig. S1), which coincide with nuclear deformation (Fig. 2A,B) and changes to NE intermembrane spacing (Fig. 2F,G). These cytoskeletal rearrangements in fibroblasts mirror those in breast cancer MDA-MB-231 cells (Fig. S1B). As paclitaxel-treated cells lack prominent actin stress fibres adjacent to the nucleus (Rosenblum and Shivers, 2000) (Fig. 1D), nuclear deformation likely occurs via the microtubule and vimentin cytoskeletons. Indeed, microtubules connect to LINC complexes in the NE via interactions between the microtubule motors kinesin and dynein and the N-terminus of KASH proteins (Wilson and Holzbaur, 2015; Gonçalves et al., 2020), whereas vimentin filaments connect to KASH proteins via plectin (Wilhelmsen et al., 2005). Furthermore, perinuclear assemblies of both microtubules and vimentin filaments have been previously observed to induce nuclear deformation in interphase (Gerlitz et al., 2013; Pan and Chen, 2024). The large, dense filament bundles in paclitaxel-treated cells might exert force on the nucleus either directly via steric effects or via their mechanical connection to LINC complexes (Wilhelmsen et al., 2005; Wilson and Holzbaur, 2015; Gonçalves et al., 2020).

LINC complexes in turn connect to the nuclear lamina in the nucleoplasm (Crisp et al., 2006; Haque et al., 2006). Here, we show that paclitaxel treatment decreases lamin A/C expression levels independently of cell division (Fig. 3B,C; Fig. 4A,B). In other systems, lamin A/C levels respond to mechanical cues (Swift et al., 2013), and this mechanosensing is thought to be achieved through post-translational modifications of lamin A/C, including phosphorylation, acetylation and ubiquitylation (Cho et al., 2017; Karoutas et al., 2019; Blank, 2020). In particular, lamin A/C phosphorylation and acetylation regulate its assembly and stability in the nuclear lamina and its subsequent degradation (Bertacchini et al., 2013; Kochin et al., 2014; Karoutas et al., 2019). Nevertheless, our results did not show any significant changes in phosphorylation, acetylation or ubiquitylation of lamin A/C upon paclitaxel treatment (Fig. S3C–H). Instead, we postulate a different mechanism involving the direct connection of lamin A/C to SUN2-containing LINC complexes (Fig. 4D,E). The role of LINC complexes in paclitaxel-induced nuclear lamina perturbations is further supported by the fact that lamin A/C but not lamin B1 levels are decreased (Fig. 3B,C). This is because, in contrast to robustly expressed lamin B1, which is not mechanically linked to the cytoskeleton directly via LINC complexes (Haque et al., 2006; Donnaloja et al., 2020), lamin A/C directly interacts with LINC complexes (Crisp et al., 2006; Haque et al., 2006), and its expression and turnover are known to be controlled by tissue stiffness and mechanical stress (Swift et al., 2013; Donnaloja et al., 2020).

A role for LINC complexes in paclitaxel-induced nuclear deformation is further supported by the uneven localisation and decreased protein levels of SUN2 (Fig. 3D–G). This aberrant SUN2 localisation in the NE likely also accounts for the altered nuclear membrane spacing upon paclitaxel treatment (Fig. 2F,G; Fig. S2K). As SUN proteins in LINC complexes contain perinuclear coiled-coil domains of a set length, their presence in the NE controls the inter-membrane distance within narrow limits (Crisp et al., 2006; Sosa et al., 2012). This control is lost when SUN proteins are absent (Crisp et al., 2006), enabling the two membranes to be more easily forced together or separated under cytoskeletal forces, as observed here (Fig. 2F,G). SUN1 is still present in paclitaxel-treated cells, and this likely explains why a large proportion of the NE still has spacing within the expected range of 20–40 nm (Fig. 2F). However, its

expression levels remain the same and therefore SUN1 is unlikely to be able to completely compensate for the reduced levels of SUN2, explaining the increased variability in NE spacing in paclitaxel-treated cells (Fig. 2F).

It is still unclear why SUN2 and not SUN1 protein levels are specifically disrupted by paclitaxel treatment (Fig. 3F,G). However, mechanical perturbations in the form of cyclic tensile strain have previously been shown to decrease levels of SUN2, but not SUN1, in human mesenchymal stem cells (Gilbert et al., 2019). We show that this decrease in SUN2 levels occurs independently of lamin A/C (Fig. S4B,C). Instead, paclitaxel treatment resulted in a significant increase in SUN2 ubiquitylation (Fig. 4F,G). This supports previous observations that the turnover of SUN2 via ubiquitylation by the E3 ubiquitin ligase SCF$^{\beta TrCP}$ is important in the control of nuclear mechanics, in particular nuclear shape (Krshnan et al., 2022). SUN1 is not under the same control given that it lacks the SCF$^{\beta TrCP}$ recognition site present in SUN2 (Krshnan et al., 2022).

Although both SUN1 and SUN2 have been shown to interact directly with lamin A/C (Crisp et al., 2006), upon paclitaxel treatment, SUN2 and lamin A/C colocalised well whereas SUN1 did not (Fig. 3D,E; Fig. S3I,J). This specific colocalisation indicates that SUN2 might be the main mediator of mechanical connections between lamin A/C and microtubules and/or vimentin filaments in human fibroblasts. We propose that these mechanical connections are what drive perturbations to lamin A/C upon paclitaxel treatment. Supporting this, the paclitaxel-induced decrease of lamin A/C protein levels was no longer observed when SUN2 was knocked down (Fig. 4D,E). This suggests that in the absence of SUN2-containing LINC complexes, aberrant cytoskeletal forces are no longer effectively transmitted to lamin A/C to disrupt its protein levels at the nuclear lamina.

Furthermore, microtubule bundling induced by GFP–Tau overexpression resulted in similar perturbations to both lamin A/C localisation and overall nuclear shape when these bundles were localised near the nucleus (Fig. 5). This confirms that these effects of paclitaxel are due to its activity on the microtubule cytoskeleton in interphase, especially because GFP–Tau overexpression did not lead to multimicronucleation or appear to affect mitosis. Furthermore, this agrees with previous studies in neuroblastoma cells showing that Tau overexpression results in nuclear deformation via microtubule bundling (Monroy-Ramírez et al., 2013).

The disruption to the nuclear lamina and NE observed upon paclitaxel treatment, caused both by aberrant nuclear–cytoskeletal coupling during interphase and following exit from defective mitosis, likely compromises the ability of the nucleus to buffer cytoskeletal forces, resulting in loss of nuclear integrity, widespread DNA damage and, if severe enough, cell death (De Vos et al., 2011; Vargas et al., 2012; Raab et al., 2016; Cho et al., 2019). Consequently, cells already containing a defective nuclear lamina or NE might be more susceptible to these effects. Indeed, we showed that lamin A/C knockdown increases the severity of nuclear deformation in paclitaxel (Fig. 2D,E) as well as the frequency of multimicronucleation (Fig. S2J), and results in increased cell death, decreased cell growth and decreased recovery from paclitaxel (Fig. 6). As nuclear abnormalities are common across most cancer types (Bell and Lammerding, 2016; Abel et al., 2024), particularly alterations to the nuclear lamina and decreased lamin A/C levels (Wu et al., 2009; Belt et al., 2011; Gong et al., 2015; Matsumoto et al., 2015; Bell and Lammerding, 2016), this study provides an explanation for the selective targeting of cancer cells by paclitaxel during interphase.

Journal of Cell Science

Overall, our work builds on previous studies investigating loss of nuclear integrity as an anti-cancer mechanism of paclitaxel action separate from mitotic arrest (Smith et al., 2021, 2022; Smith and Xu, 2021). We propose that cancer cells show increased sensitivity to nuclear deformation induced by aberrant nuclear–cytoskeletal coupling and multimicronucleation following mitotic slippage. Therefore, we conclude that paclitaxel functions in interphase as well as mitosis, elucidating how slowly growing tumours are targeted (Milross et al., 1996; Orth et al., 2011; Janssen et al., 2013). Given that better understanding the mechanisms of paclitaxel and its effects on nuclear architecture is important for its safe and efficacious use as an anti-cancer drug, our study might inform patient stratification and the development of safer drug formulations. Similarly, our findings might be relevant for the search and/or design of other drugs that affect nuclear–cytoskeletal coupling.

## MATERIALS AND METHODS

### Cell culture and paclitaxel treatment

Human skin fibroblasts derived from AG10803 and immortalised with SV40LT and TERT were a gift from Dr Delphine Larrieu (Altos Labs, Cambridge Institute of Science, UK). MDA-MB-231 human breast cancer cells were purchased from Merck. Cells were cultured at 37°C and 5% $CO_2$ in complete medium consisting of Dulbecco's modified Eagle's medium (DMEM) with GlutaMAX (Gibco), supplemented with 10% fetal bovine serum (FBS) and 1% penicillin-streptomycin (Gibco; denoted as complete medium). Cells in culture were tested negative for the presence of mycoplasma.

For serum starvation, cells were resuspended in PBS before being pelleted at 300 $g$ for 5 min at room temperature. Cells were then washed in PBS twice, before being plated in low-serum medium [minimal essential medium (MEM; Gibco) with 0.5% FBS and 1% penicillin-streptomycin]. Cells were incubated in low-serum medium for 3 days prior to further experiments.

For paclitaxel treatment, paclitaxel (Merck PHL89806) resuspended in dimethyl sulfoxide (DMSO) to 10 mM was diluted in complete DMEM to final concentrations of 1, 5 or 10 nM as indicated.

### Transfections and RNA interference

To overexpress lamin A, cells were transfected using Lipofectamine 3000 (Invitrogen) with pBABE-puro-GFP-lamin A plasmid (Addgene plasmid 17662, deposited by Tom Misteli; Scaffidi and Misteli, 2008). To overexpress Tau, cells were electroporated using a 4D nucleofector X kit (Lonza) with pRK5 GFP-Tau plasmid (Addgene plasmid 187023, deposited by George Bloom; Rajbanshi et al., 2023). Overexpression was confirmed by fluorescence microscopy (Fig. 2D; Fig. 5A).

For knockdowns, cells were transfected with the indicated small interfering RNA (siRNA) at a final concentration of 100 nM using Lipofectamine 3000 (Table S1). Knockdown was confirmed by western blot analysis (Fig. S2E; Fig. S4A). Drug treatments were started 48 h after transfection.

### High-content live-cell imaging

Cells adhered to 24-well plates (Corning) were stained for 10 min with Hoechst 33342 (Thermo Fisher Scientific). The indicated concentration of paclitaxel was then added, and 24-well plates were transferred to CELLCYTE X Live Cell Imager (Cytena) fitted with a 10× objective lens (air, NA 0.3) held at 37°C and 5% $CO_2$ and imaged every 2 h. For paclitaxel recovery, wells were washed and the medium replaced with complete medium after 24 h.

To quantify nuclear shape deformation, images were transferred to ImageJ for analysis; Hoechst-stained nuclei were selected by threshold and the 'analyse particles' function used to calculate nuclear solidity. Nuclei that were overlapping with those from adjacent cells or that were within dead cells, identified by their distinctive cell shape, were excluded from the analysis.

Cell growth was quantified by measuring confluency from brightfield images using the automated image analysis tool in CELLCYTE X (Cytena).

### Cell staining and immunofluorescence

Following transfection and/or paclitaxel treatment, cells adhered on #1.5 glass coverslips were stained with Hoechst 33342 (Invitrogen) in complete medium for 10 min at 37°C. Cells were then washed with PBS, fixed with 4% paraformaldehyde for 15 min at room temperature, then washed with PBS before autofluorescence was quenched using 50 mM ammonium chloride. Cells were then permeabilised and blocked in permeabilisation buffer (0.1% Triton X-100 and 2% bovine serum albumin in PBS) for 30 min then washed in PBS. Cells were then incubated at room temperature for 1 h in permeabilisation buffer with the indicated primary antibodies (Table S2) or Alexa Fluor 488–phalloidin (Invitrogen). After washing with PBS, cells were incubated at room temperature for 1 h in secondary antibodies in permeabilisation buffer (Table S2) then washed with PBS. For STORM samples, cells were then incubated with 4% paraformaldehyde for 10 min then washed and stored in PBS. Otherwise, cells were mounted using Prolong Gold (Thermo Fisher Scientific).

For quantification of cell viability, cells adhered on #1.5 glass coverslips pre-stained with Hoechst 33342 were then stained for active caspases 3 and 7 using FLICA reagent from the Image-iT LIVE Red caspase detection kit (Invitrogen). Cells were then washed twice, fixed with 4% paraformaldehyde for 15 min at room temperature, washed with PBS, then mounted. Following confocal microscopy, the proportion of dead or apoptotic cells was calculated by measuring the proportion of cells that were stained using FLICA.

### Light microscopy

Immunofluorescence imaging was performed using NIS-Elements software and a Nikon X1 Spinning Disk inverted microscope equipped with a 40× objective lens [oil immersion, numerical aperture (NA) 1.3] and an sCMOS camera.

For stochastic optical reconstruction microscopy (STORM), immunofluorescence samples were immersed in STORM buffer (10% glucose, 10 mM sodium chloride, 50 mM Tris-HCl pH 8.0, 5.6% glucose oxidase, 3.4 mg/ml catalase and 0.1% 2-mercaptoethanol) and imaged using a Nanoimager system (ONI) equipped with a 100× TIRF objective lens (oil immersion, NA 1.49). For imaging, highly inclined and laminated optical sheet (HILO) illumination was used with laser power 60–150 mW and 15,000 frames were acquired. CODI software (ONI) was used for image processing, including drift correction and HDBSCAN clustering. For α-tubulin cluster analysis, HDBSCAN clusters were constrained to have a minimum number of 15 localisations, maximum circularity of 0.5 and minimum length of 1000 nm, so that the clusters mapped to individual filament bundles.

### Pull-downs and western blotting

For western blotting, cells were collected by treatment with 0.25% trypsin-EDTA for 3 min then dilution in PBS followed by centrifugation at 400 $g$ for 10 min. Cells were lysed with 1× NUPAGE LDS sample buffer supplemented with 50 mM dithiothreitol (DTT). After boiling at 95°C for 10 min, proteins were separated using NUPAGE 4–12% Bis-Tris gels (Thermo Fisher Scientific) in MES SDS buffer alongside a protein ladder (Invitrogen SeeBlue Plus2 or Bio-Rad Precision Plus All Blue Protein Standards). Proteins were transferred to PVDF membranes (Bio-Rad) which were then incubated in blocking buffer [5% milk power and 0.1% Tween 20 in Tris-buffered saline (TBS)]. The membranes were then incubated in primary antibodies in blocking buffer (Table S2), then washed with washing buffer (1% milk and 0.1% Tween 20 in TBS), then incubated with secondary antibodies in blocking buffer (Table S2), then washed in washing buffer again. The membranes were then activated using ECL (Cytiva) then imaged using a Bio-Rad Chemidoc MP. Relative protein levels were quantified in ImageJ from three biological repeats, with normalisation to Cyclophilin B loading controls.

For Phos-tag gel analysis, cells were washed in PBS then lysed in cold lysis buffer [10 mM HEPES pH 7.5, 2 mM magnesium chloride, 25 mM potassium chloride, 0.1% NP-40, 0.1% Triton X-100, 0.1 mM DTT, 0.1 mM PMSF, and cOmplete EDTA-free inhibitor cocktail (Sigma-Aldrich)] with cell scraping. Following incubation on ice for 1 h, cell lysates were centrifuged at 14,000 $g$ for 15 min at 4°C, then the supernatant

was boiled in 1× NUPAGE LDS sample buffer with 50 mM DTT before separation using 50 μM Phos-tag 10% acrylamide gels (FUJIFILM) in Tris-glycine SDS buffer. After running, the gel was washed twice in Tris-glycine SDS buffer with 1 mM EDTA, then once in Tris-glycine SDS buffer. Proteins were then transferred to PVDF membranes for western blot analysis as above.

For pull-downs, cells were lysed as for the Phos-tag gel analysis. Cell lysates were centrifuged at 14,000 *g* for 15 min at 4°C, before 250 μl of the lysate supernatant was pre-cleared by incubating with 25 μl protein A Dynabeads (Invitrogen) for 20 min at room temperature with end-to-end rotation. The beads were then magnetically separated and the pre-cleared lysate incubated with primary antibodies (Table S2) overnight at 4°C with end-to-end rotation. The lysate–antibody mix was then added to 50 μl protein A Dynabeads and incubated for 1 h at room temperature with end-to-end rotation. The lysate was then removed by magnetic separation, and the beads washed three times in cold lysis buffer. The pulled-down proteins were then eluted by boiling the beads in 50 μl 1× NUPAGE LDS sample buffer for 5 min, before the beads were magnetically removed. Samples were then separated by gel electrophoresis for western blotting as above. For loading controls, the membranes were then stripped by incubation in stripping buffer (15 mg/ml glycine, 1 mg/ml SDS and 1% Tween 20 in water, adjusted to pH 2.2) for 10 min twice, before washing them twice in PBS for 10 min then twice in TBS with 0.1% Tween 20 for 5 min. The membranes were then blocked and probed with either anti-lamin A/C or anti-SUN2 antibodies as above. Uncropped scans of western blots are show in Fig. S6.

### Reverse transcription quantitative PCR

To extract mRNA, cells were washed with ice-cold PBS then lysed with 1 ml TRIzol (Invitrogen) for 2 min at room temperature, before shaking with 250 μl chloroform. After 5 min, the mixture was centrifuged at 10,000 *g*. The top aqueous layer was then removed, added to 550 μl isopropanol, incubated at room temperature for 5 min, then centrifuged at 14,000 *g* for 30 min. The tube was placed on ice and the isopropanol poured off, before adding 1 ml 75% ethanol. After centrifugation at 9500 *g* for 5 min, the ethanol was poured off before the RNA pellets were dried then resuspended in water. For reverse transcription, the TaqMan reverse transcriptase kit (Thermo Fisher Scientific) was used with random hexamer primers according to the manufacturer's protocol. For qPCR, 0.6 μl cDNA was added to 3.4 μl water, 5 μl iTaq Universal SYBR Green Supermix (Bio-Rad), 0.5 μl 0.5 μM forward primers, and 0.5 μl 0.5 μM reverse primers (Table S3). Three biological repeats were performed for the control and paclitaxel-treated samples. For each of these biological repeats, three technical replicates were performed using primers targeting the gene of interest, and three using primers targeting the housekeeping gene GAPDH. Forty cycles with denaturation at 95°C for 10 s and annealing-extension at 60°C for 60 s were run in a Bio-Rad CFX RT-qPCR machine. The cycle threshold (Ct) values were then used for $2^{-\Delta\Delta Ct}$ analysis, as previously described (Livak and Schmittgen, 2001).

### EM grid preparation

Quantifoil R 1/4 Au 200 mesh grids (Quantifoil Micro Tools GmbH) were glow discharged for 40 s at 30 mA with an Edwards S150B glow discharger then coated using 0.1 mg/ml poly-L-lysine for 30 min. Grids were then washed in PBS three times, then cells were seeded and allowed to adhere overnight, before 16 h treatment with 5 nM paclitaxel or control medium. Grids were then transferred to a Leica GP2 plunger at 25°C and 95% humidity using glycerol (10% in PBS) as a cryo-protectant for 20 s. After 9 s blotting, grids were plunge frozen in liquid ethane.

### Cryo-focussed ion beam milling and scanning electron microscopy

To prepare lamellae for cryo-ET acquisition, cryo-FIB milling was performed using either an Aquilos-II (Thermo Fisher Scientific) or Scios dual beam focussed ion beam milling and scanning electron microscope (FIB and SEM; Thermo Fisher Scientific) equipped with a PP3010 T cryo stage and loading system (Quorum Technologies) according to previously published protocols (Wagner et al., 2020). Briefly, the grids were sputter-coated with an inorganic platinum layer at 10 mA, then coated with an

organometallic platinum layer {trimethyl [(1,2,3,4,5-ETA)-1 methyl-2, 4-cyclopentadien-1-YL] platinum} using the gas injection system. Milling was conducted at an angle of 10° relative to the grid in a stepwise manner with decreasing currents (0.5 nA, 0.3 nA, 0.1 nA and 30 pA), resulting in lamellae with a final width of 10–12 μm and thickness of 120–200 nm.

### Electron cryo-tomography and morphometrics analysis

Lamellae were imaged using a Titan Krios (TFS) equipped with a Gatan BioQuantum energy-filter and Gatan K3 direct electron detector. Tilt series were collected using SerialEM (Mastronarde, 2005) with PACE-tomo (Eisenstein et al., 2023) in a dose-symmetric scheme between±60° relative to the lamella in 3° increments. The total dose for each tilt series was ∼100 e/Å², distributed evenly across each 12-frame tilt. The pixel size was 1.63 Å/pixel and defocus −3 to −5 μm.

For tomogram reconstruction, movies were imported into Warp (Tegunov and Cramer, 2019) for gain and motion correction, tilt selection and CTF estimation. Tomograms were then reconstructed and deconvolved following tilt series alignment using AreTomo (Zheng et al., 2022).

Microtubules, vimentin filaments, and the inner and outer nuclear membranes were manually segmented in IMOD (Kremer et al., 1996). To analyse NE morphology, the NE segmentations were then converted into 3D voxel segmentations using the 'imodmop' program within IMOD. A surface morphometrics pipeline (Barad et al., 2023) was then used to generate surface meshes of the inner and outer NEs from the 3D voxel segmentations and subsequently measure the distances between these meshes, giving the INM to ONM distance.

### Statistical analysis

Statistical analyses were performed using Prism 10 (GraphPad), Python (*scipy.stats*; https://docs.scipy.org/doc/scipy/reference/stats.html) and Origin2022 (OriginLabs). Unpaired two-tailed Student's *t*-tests were used to compare data between two conditions. For western blot quantifications, band intensities were first normalized to their respective loading controls and subsequently to the control condition (0 nM paclitaxel). One-sample two-tailed *t*-tests (null hypothesis mean=1) were then used to assess deviations relative to the control condition. To evaluate differences in the distribution of INM–ONM distances, Levene's test for equality of variances was performed using the *scipy.stats* module in Python. The effects of lamin A/C expression levels on cell growth over time in the presence of paclitaxel were analysed using one-way three-factor repeated-measures ANOVAs followed by Bonferroni post hoc tests in Origin2022. A significance threshold of $P<0.05$ was applied to all analyses. The statistical tests and corresponding *P*-values are reported in the figure legends.

### Acknowledgements

We thank Dr David Barford, Dr Andrew Carter and Prof. Laura Machesky for in-depth discussions and critical feedback. We thank Dr Delphine Larrieu for providing hTERT human skin fibroblasts used in this study. We thank the EM Facility at the LMB, the LMB Scientific Computing facility and the LMB Light Microscopy Facility for technical support. We also acknowledge the UK national electron Bio-Imaging Centre (eBIC) for access to Aquilos-II.

### Competing interests

The authors declare no competing or financial interests.

### Author contributions

Conceptualization: T.H., A.d.S., M.A.; Data curation: T.H., V.L.H., P.K., A.d.S., M.A.; Formal analysis: T.H., V.L.H.; Funding acquisition: M.A.; Investigation: T.H.; Methodology: T.H., V.L.H., P.K., A.d.S., M.A.; Project administration: A.d.S., M.A.; Resources: M.A.; Supervision: V.L.H., P.K., A.d.S., M.A.; Validation: V.L.H., A.d.S., M.A.; Visualization: T.H.; Writing – original draft: T.H.; Writing – review & editing: V.L.H., P.K., A.d.S., M.A.

### Funding

This work was supported by the Medical Research Council, as part of United Kingdom Research and Innovation (also known as UK Research and Innovation) [MC_UP_1201/30]. M.A. was funded by the UKRI Medical Research Council (MC_UP_1201/30), A.d.S. was funded by the Wellcome Trust Early Career Award to A.d.S. (227622/Z/23/Z). Open Access funding provided by Medical Research Council Laboratory of Molecular Biology. Deposited in PMC for immediate release.

## Data and resource availability

All relevant data and details of resources can be found within the article and its supplementary information.

## Peer review history

The peer review history is available online at https://journals.biologists.com/jcs/lookup/doi/10.1242/jcs.264494.reviewer-comments.pdf

## Special Issue

This article is part of the Special Issue 'Cell Biology of the Nucleus', guest edited by Abby Buchwalter. See related articles at https://journals.biologists.com/jcs/issue/139/12.

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
