## [Peer Review File · Journal of Cell Science]

Paclitaxel compromises nuclear integrity in interphase through SUN2-mediated cytoskeletal coupling

Thomas Hale, Victoria Louise Hale, Piotr Kolata, Alia dos Santos and Matteo Allegretti
DOI: 10.1242/jcs.264494

Editor: Megan King

Review timeline

Submission to Review Commons:	30 January 2025
Submission to Journal of Cell Science:	7 October 2025
Editorial Decision:	20 October 2025
First revision received:	10 November 2025
Accepted:	17 November 2025

Reviewer 1

Evidence, reproducibility and clarity

This description of the down-regulation of the expression of lamin A/C upon treatment with paclitaxel and its sensitivity to SUN2 is quite interesting but still somehow preliminary. It is unclear whether this effect involves the regulation of gene expression, or of the stability of the proteins. How SUN2 mediates this effect is still unknown. The roles of free tubulins and polymerized microtubules, and thus the potential role of paclitaxel, need to be uncovered.

The doses of paclitaxel at which occur the effects described in the paper are not fully consistent with all the conclusions. Most experiments have been done at 5 nM. However, at this dose the effect of lamin A/C over or down expression on the growth (differences in the slopes of the curves in Figure 4A) are not fully convincing and not fully consistent with the clear effect on viability as well (in addition, duration of treatments before assessing viability are not specified). At 1 nM, cell growth is reduced and the rescuing effect of lamin over-expression is much more clear (Fig 4A), and the nucleus deformation clear (Fig 2A) but this dose has no effect on lamin A/C expression (Fig 3C), which questions how lamins impact nucleus shape and cell survival. Cytoskeleton reorganisation in these conditions is not described although it could clarify the respective role of force production (suggested in figure 1) and nuclei resistance (shown in figure 2) in paclitaxel sensitivity.

Finally, although the absence of role of mitotic arrest is clear from the data, the defective reorganisation of the nucleus after mitosis still suggest that the effect of paclitaxel is not independent of mitosis.

minor: a more thorough introduction of known data about dose response of cells in culture and in vivo would help understanding the range of concentrations used in this study.

Significance

In this manuscript, Hale and colleagues describe the effect of paclitaxel on nucleus deformation and cell survival. They showed that 5nM of paclitaxel induces nucleus fragmentation, cytoskeleton reorganisation, reduced expression of LaminA/C and SUN2, and reduced cell growth and viability. They also showed that these effects could be at least partly compensated by the over-expression of lamin A/C. As fairly acknowledged by the authors, the induction of nuclear deformation in

paclitaxel-treated cells, and the increased sensitivity to paclitaxel of cells expressing low level of lamin A/C are not novel (reference #14). Here the authors provided more details on the cytoskeleton changes and nuclear membrane deformation upon paclitaxel treatment. The effect of lamin A/C over and down expression on cell growth and survival are not fully convincing, as further discussed below. The most novel part is the observation that paclitaxel can induce the down-regulation of the expression of lamin A/C and that this effect is mediated by SUN2.

Reviewer 2

Evidence, reproducibility and clarity

This study investigates the effects of the chemotherapeutic drug paclitaxel on nuclear-cytoskeletal coupling during interphase, claiming a novel mechanism for its anti-cancer activity. The study uses hTERT-immortalized human fibroblasts. After paclitaxel exposure, a suite of state-of-the-art imaging modalities visualizes changes in the cytoskeleton and nuclear architecture. These include STORM imaging and a large number of FIB-SEM tomograms.

Major comments:

The authors make a major claim that in addition to the somewhat well-described mechanism of paclitaxel on mitosis, they have discovered 'an alternative, poorly characterised mechanism in interphase'.

However, none of the data proves that the effects shown are independent of mitosis. To the contrary, measurements are presented 48 hours after paclitaxel treatment starts, after which it can be assumed that 100% of cells have completed at least one mitotic event. The appearance of micronuclei evidences this, as discussed by the authors shortly.

It looks like most of the results shown are based on botched mitosis or, more specifically, errors on nuclear assembly upon exit from mitosis rather than a specific effect of paclitaxel on interphase. The readouts the authors show just happen to be measurements while the cells are in interphase.

Alternative hypotheses are missing throughout the manuscript, and so are critical controls and interpretations.

The authors claim that 'Previously, the anti-cancer activity of paclitaxel was thought to rely mostly on the activation of the mitotic checkpoint through disruption of microtubule dynamics, ultimately resulting in apoptosis.'

The authors may have overlooked much of the existing literature on the topic, including many recent manuscripts from Xiang-Xi Xu's and another lab.

The data, e.g. in Figure 1, does not hold up to the first alternative hypothesis, e.g. that paclitaxel stabilizes microtubules and that excessive mechanical bundling of microtubules induces major changes to cell shape and mechanical stress on the nucleus.

Even the simplest controls for this effect (the application of an alternative MT stabilizing drug or the overexpression of an MT stabilizer, e.g., tau).

The focus on nuclear lamina seems somewhat arbitrary and adjacent to previously published work by other groups.

What would happen if the authors stained for focal adhesion markers? There would probably be a major change in number and distribution. Would the authors conclude that paclitaxel exerts a specific effect on focal adhesions? Or would the conclusion be that microtubule stabilization and the following mechanical disruption induce pleiotropic effects in cells?

Which effects are significant for paclitaxel function on cancer cells?

Minor comments:

While I understand the difficulty of the experiments and the effort the authors have put into producing FIB-SEM tomograms, I am not sure they are helping their study or adding anything beyond the light microscopy images. Some of the images may even be in the way, such as

supplementary Figure 6, which lacks in quality, controls, and interpretation. Do I see a lot of mitochondria in that slice?

I may have overlooked it, but has the number of cells from which lamellae have been produced been stated?

Significance

The significance of studying the effect of paclitaxel, the most successful chemotherapy drug, should be broad and of interest to basic researchers and clinicians.

As outlined above, I believe that major concerns about the design and interpretation of the study hamper its significance and advancements.

My areas of expertise could be broadly defined as Cell Biology, Cytoskeleton, Microtubules, and Structural Biology.

Reviewer 3

Evidence, reproducibility and clarity

The manuscript presents interesting new ideas for the mechanism of an old drug, taxol, which has been studied for the last 40 years. Although similar ideas are published, which may be suitable to be cited?

- Paclitaxel resistance related to nuclear envelope structural sturdiness.

Smith ER, Wang JQ, Yang DH, Xu XX. Drug Resist Updat. 2022 Dec;65:100881. doi: 10.1016/j.drug.2022.100881. Epub 2022 Oct 15. PMID: 36368286 Review.

- Breaking malignant nuclei as a non-mitotic mechanism of taxol/paclitaxel.

Smith ER, Xu XX. J Cancer Biol. 2021;2(4):86-93. doi: 10.46439/cancerbiology.2.031. PMID: 35048083 Free PMC article.

Only one cell line was used in all the experiments? "Human telomerase reverse transcriptase (hTERT) immortalised human fibroblasts" ? The cells used are not very relevant to cancer cells (carcinomas) that are treated with paclitaxel. It is not clear if the observations and conclusions will be able to be generalized to cancer cells.

"Fig. 1. (B) STORM imaging of α -tubulin immunofluorescence in cells fixed after 16 h incubation in control media or 5 nM paclitaxel. Lower panels show α -tubulin clusters generated with HDBSCAN analysis. Scale bars = 10 μ m."

It needs explanation of what is meaning of the different color lines in the lower panels, just different filaments?

Generally, the figures need additional description to be clear.

"Figure 3 - Paclitaxel results in aberrations to the nuclear lamina." The sentence seems not to be well constructed. "Paclitaxel treatment causes ..."?

Lamin A and C levels are different in different images (Fig. 3B, H): some Lamin A is higher, and sometime Lamin C is higher? This may possibly due to culture condition or subtle difference in sample handling?. Also, the effect on Lamin A/C and SUN2 levels are not significant or robust. Any mechanisms are speculated for the reason for the reduction? Also, the about 50% reduction in protein level is difficult to be convincing as an explanation of nuclear disruption.

Significance

The manuscript presents interesting new ideas for the mechanism of an old drug, taxol, which has been studied for the last 40 years.

The data may be improved to provide stronger support.

Additional cell lines (of cancer or epithelial origin) may be repeated to confirm the generality of the observation and conclusions.?

Author response to reviewers' comments

Manuscript number: RC-2025-02879

Corresponding author(s): Matteo Allegretti; Alia dos Santos

1. General Statements [optional]

This section is optional. Insert here any general statements you wish to make about the goal of the study or about the reviews.

In this study, we investigated the effects of paclitaxel on both healthy and cancerous cells, focusing on alterations in nuclear architecture. Our novel findings show that:

- Paclitaxel-induced microtubule reorganisation during interphase alters the perinuclear distribution of actin and vimentin. The formation of extensive microtubule bundles, in paclitaxel or following GFP-Tau overexpression, coincides with nuclear shape deformation, loss of regulation of nuclear envelope spacing, and alteration of the nuclear lamina.
- Paclitaxel treatment reduces Lamin A/C protein levels via a SUN2-dependent mechanism. SUN2, which links the lamina to the cytoskeleton, undergoes ubiquitination and consequent degradation following paclitaxel exposure.
- Lamin A/C expression, frequently dysregulated in cancer cells, is a key determinant of cellular sensitivity to, and recovery from, paclitaxel treatment.

Collectively, our data support a model in which paclitaxel disrupts nuclear architecture through two mechanisms: (i) aberrant nuclear-cytoskeletal coupling during interphase, and (ii) multimicronucleation following defective mitotic exit. This represents an additional mode of action for paclitaxel beyond its well-established mechanism of mitotic arrest.

We thank the reviewers for their time and constructive feedback. We have carefully considered all comments and have carried out a full revision. The updated manuscript now includes additional data showing:

- § Overexpression of microtubule-associated protein Tau causes similar nuclear aberration phenotypes to paclitaxel. This supports our hypothesis that increased microtubule bundling directly leads to nuclear disruption in paclitaxel during interphase.
- § Paclitaxel's effects on nuclear shape and Lamin A/C and SUN2 expression levels occur independently of cell division.
- § Reduced levels of Lamin A/C and SUN2 upon paclitaxel treatment occur at the protein level via ubiquitination of SUN2.
- § The effects of paclitaxel on the nucleus are conserved in breast cancer cells.

We have also edited our text and added further detail to clarify points raised by the reviewers. We believe that our revised manuscript is overall more complete, solid and compelling thanks to the reviewers' comments.

This section is mandatory. Please insert a point-by-point reply describing the revisions that were already carried out and included in the transferred manuscript.

Reviewer #1**Evidence, reproducibility and clarity**

This description of the down-regulation of the expression of lamin A/C upon treatment with paclitaxel and its sensitivity to SUN2 is quite interesting but still somehow preliminary. It is unclear whether this effect involves the regulation of gene expression, or of the stability of the proteins. How SUN2 mediates this effect is still unknown.

We thank the reviewer for this valuable comment. To elucidate the mechanism behind the decrease in Lamin A/C and SUN2 levels, we have now performed several additional experiments. First, we performed RT-qPCR to quantify mRNA levels of these genes, relative to the housekeeping gene *GAPDH* (**Supplementary Figure 3B and O**). The levels of *SUN2* and *LMNA* mRNA remained the same between control and paclitaxel-treated cells, indicating that this effect instead occurs at the protein level. We have also tested post-translational modifications as a potential regulatory mechanism for Lamin A/C and SUN2. In addition to the phosphorylation of Ser404 which we had already tested (**Supplementary Figure 3C**), we have now included additional Phos-tag gel and Western blotting data showing that the overall phosphorylation status of Lamin A/C is not affected by paclitaxel (**Supplementary Figure 3E and F**). We also pulled-down Lamin A/C from cell lysates and then Western blotted for polyubiquitin and acetyl-lysine, which showed that the ubiquitination and acetylation states of Lamin A/C are also not affected by paclitaxel (**Supplementary Figure 3G-I**). However, Western blots for polyubiquitin of SUN2 pulled down from cell lysates showed that paclitaxel treatment results in significant SUN2 ubiquitination (**Figure 3M and N**). Therefore, we propose that the downregulation of SUN2 following paclitaxel treatment occurs by ubiquitin-mediated proteolysis.

The roles of free tubulins and polymerized microtubules, and thus the potential role of paclitaxel, need to be uncovered.

We addressed this important point by using an alternative method to stabilise/bundle microtubules in interphase, namely by overexpressing GFP-Tau, as suggested by reviewer 2. Following GFP-Tau overexpression, large microtubule bundles were observed throughout the cytoplasm (**Figure 4A**), and this resulted in a significant decrease in nuclear solidity (**Figure 4B**). Furthermore, in cells where microtubule bundles extensively contacted the nucleus, the nuclear lamina became unevenly distributed and appeared patchy (**Figure 4C**). This supports our hypothesis that the aberrations to nuclear shape and Lamin A/C localisation in paclitaxel-treated cells are due to the presence of microtubules bundles surrounding the nucleus.

The doses of paclitaxel at which occur the effects described in the paper are not fully consistent with all the conclusions. Most experiments have been done at 5 nM. However, at this dose the effect of lamin A/C over or down expression on the growth (differences in the slopes of the curves in **Figure 4A**) are not fully convincing and not fully consistent with the clear effect on viability as well (in addition, duration of treatments before assessing viability are not specified). At 1 nM, cell growth is reduced and the rescuing effect of lamin over-expression is much more clear (**Fig 4A**), and the nucleus deformation clear (**Fig 2A**) but this dose has no effect on lamin A/C expression (**Fig 3C**), which questions how lamins impact nucleus shape and cell survival. Cytoskeleton reorganisation in these conditions is not described although it could clarify the respective role of force production (suggested in **figure 1**) and nuclei resistance (shown in **figure 2**) in paclitaxel sensitivity.

We thank the reviewer for raising this important point. We have addressed this by conducting additional repeats for the cell confluency measurements to increase the statistical power of our experiments (**Figure 5A**). Our data now show that GFP-lamin A/C had a statistically significant effect on rescuing cell growth at both 1 nM and 5 nM paclitaxel, while Lamin A/C knockdown exacerbated the inhibition of cell growth at 5 nM paclitaxel but not 1 nM paclitaxel (**Figure 5A**). In addition, we note that the duration of paclitaxel treatment before assessing viability was specified in the figure legend: “*Bar graph comparing cell viability between wild-type (red), GFP-Lamin A/C overexpression (green), and Lamin A/C knockdown (blue) cells following 20 h incubation in 0, 1, 5, or 10 nM paclitaxel.*” We also repeated cell viability analysis after 48 h incubation in paclitaxel instead of 20 h to allow for a longer time for differences to take effect (**Figure 5B**).

We also added figures showing the cytoskeletal reorganisation at both 1 and 10 nM in addition to 0 and 5 nM (**Supplementary Figure 1A**) showing that microtubule bundling and condensation of actin into puncta correlated with increased paclitaxel concentration. Vimentin colocalised well with microtubules at all concentrations.

We have also included in our results section further clarification for the use of 5nM paclitaxel in this study. The new section reads as follows: *“Experiments were performed at 5 nM paclitaxel (with additional experiments to determine dose relationships at 1 and 10 nM) because this aligns with previous studies^{7,14,24}. Furthermore, previous analysis of patient plasma reveals that typical concentrations are within the low nanomolar range⁸, and concentrations of 5-10 nM are required in cell culture to reach the same intracellular concentrations observed in vivo in patient tumours⁹. This aligns with in vitro cytotoxic studies of paclitaxel in eight human tumour cell lines which show that paclitaxel’s IC50 ranges between 2.5 and 7.5 nM⁴¹.”*

Finally, although the absence of role of mitotic arrest is clear from the data, the defective reorganisation of the nucleus after mitosis still suggest that the effect of paclitaxel is not independent of mitosis.

We thank the reviewer for pointing out the need for clarification in the wording of our manuscript. We have reworded the title and relevant sections of our abstract, introduction, and discussion to make it clearer that the effects of paclitaxel on the nucleus are due to a combination of aberrant nuclear cytoskeletal coupling during interphase and multimicronucleation following mitotic slippage. We have also added additional data in support of the effect of paclitaxel on nuclear architecture during interphase. For this, we used serum-starved cells (which divide only very slowly such that the majority of cells do not pass through mitosis during the 16 h incubation in paclitaxel [**Supplementary Figure 2D**]). Our new data confirmed that paclitaxel’s effects on nuclear solidity, and Lamin A/C and SUN2 proteins levels can occur independently of cell division (**Figure 2C**; **Figure 3H-J**). Finally, when we overexpressed GFP-Tau (as discussed above) we observed similar aberrations to nuclear solidity and Lamin A/C localisation. This indicates that these effects occur due to microtubule bundling in interphase, especially as in our study GFP-Tau did not lead to multimicronucleation or appear to affect mitosis (**Figure 4**).

Below are the main changes to the text regarding the interphase effect of paclitaxel:

- Title: *“Paclitaxel compromises nuclear integrity in interphase through SUN2-mediated cytoskeletal coupling”*
- Abstract: *“Overall, our data supports nuclear architecture disruption, caused by both aberrant nuclear-cytoskeletal coupling during interphase and exit from defective mitosis, as an additional mechanism for paclitaxel beyond mitotic arrest.”*
- Introduction: *“Here we propose that cancer cells have increased vulnerability to paclitaxel both during interphase and following aberrant mitosis due to pre-existing defects in their NE and nuclear lamina.”*
- Discussion: *“Overall, our work builds on previous studies investigating loss of nuclear integrity as an anti-cancer mechanism of paclitaxel separate from mitotic arrest^{14,20,21}. We propose that cancer cells show increased sensitivity to nuclear deformation induced by aberrant nuclear-cytoskeletal coupling and multimicronucleation following mitotic slippage. Therefore, we conclude that paclitaxel functions in interphase as well as mitosis, elucidating how slowly growing tumours are targeted.”*

minor: a more thorough introduction of known data about dose response of cells in culture and in vivo would help understanding the range of concentrations used in this study.

As mentioned above, we have now included additional information in our Results section to clarify our paclitaxel dose range:

“Experiments were performed at 5 nM paclitaxel (with additional experiments to determine dose relationships at 1 and 10 nM) because this aligns with previous studies^{7,14,24}. Furthermore, previous analysis of patient plasma reveals that typical concentrations are within the low

nanomolar range⁸, and concentrations of 5-10 nM are required in cell culture to reach the same intracellular concentrations observed in vivo in patient tumours⁹. This aligns with in vitro cytotoxic studies of paclitaxel in eight human tumour cell lines which show that paclitaxel's IC50 ranges between 2.5 and 7.5 nM⁴¹."

Significance

In this manuscript, Hale and colleagues describe the effect of paclitaxel on nucleus deformation and cell survival. They showed that 5nM of paclitaxel induces nucleus fragmentation, cytoskeleton reorganisation, reduced expression of LaminA/C and SUN2, and reduced cell growth and viability. They also showed that these effects could be at least partly compensated by the over-expression of lamin A/C. As fairly acknowledged by the authors, the induction of nuclear deformation in paclitaxel-treated cells, and the increased sensitivity to paclitaxel of cells expressing low level of lamin A/C are not novel (reference #14). Here the authors provided more details on the cytoskeleton changes and nuclear membrane deformation upon paclitaxel treatment. The effect of lamin A/C over and down expression on cell growth and survival are not fully convincing, as further discussed below. The most novel part is the observation that paclitaxel can induce the down-regulation of the expression of lamin A/C and that this effect is mediated by SUN2.

We appreciate the reviewer's summary and thank them for their time. We believe our comprehensive revisions have addressed all comments, strengthening the manuscript and making it more robust and compelling.

Reviewer #2

Evidence, reproducibility and clarity

This study investigates the effects of the chemotherapeutic drug paclitaxel on nuclear-cytoskeletal coupling during interphase, claiming a novel mechanism for its anti-cancer activity. The study uses hTERT-immortalized human fibroblasts. After paclitaxel exposure, a suite of state-of-the-art imaging modalities visualizes changes in the cytoskeleton and nuclear architecture. These include STORM imaging and a large number of FIB-SEM tomograms.

We thank the reviewer for the summary and for highlighting our efforts in using the latest imaging technical advances.

Major comments:

The authors make a major claim that in addition to the somewhat well-described mechanism of paclitaxel on mitosis, they have discovered 'an alternative, poorly characterised mechanism in interphase'.

However, none of the data proves that the effects shown are independent of mitosis. To the contrary, measurements are presented 48 hours after paclitaxel treatment starts, after which it can be assumed that 100% of cells have completed at least one mitotic event. The appearance of micronuclei evidences this, as discussed by the authors shortly. It looks like most of the results shown are based on botched mitosis or, more specifically, errors on nuclear assembly upon exit from mitosis rather than a specific effect of paclitaxel on interphase. The readouts the authors show just happen to be measurements while the cells are in interphase.

Alternative hypotheses are missing throughout the manuscript, and so are critical controls and interpretations.

We thank the reviewer for highlighting the lack of clarity in our wording. We have revised the title, abstract and relevant sections of the introduction and discussion to clarify our message that the effects of paclitaxel on the nucleus arise from a combination of aberrant nuclear-cytoskeletal coupling during interphase and multimicronucleation following exit from defective mitosis.

We have also included additional data where we used slow-dividing, serum-starved cells (under

these conditions, the majority of cells do not undergo mitosis during the 16 h incubation in paclitaxel [Supplementary Figure 2D]). Our new data show that even in these cells there is a clear effect of paclitaxel on nuclear solidity, and Lamin A/C and SUN2 protein levels, further supporting our hypothesis that these phenotypes can occur independently of cell division (Figure 2C; Figure 3H-J).

Furthermore, we performed additional experiments where we used overexpression of GFP-Tau as an alternative method of stabilising microtubules in interphase and observed similar aberrations to nuclear solidity and Lamin A/C localisation. As GFP-Tau overexpression did not lead to micronucleation or appear to affect mitosis, these data support the hypothesis that nuclear aberrations occur due to microtubule bundling in interphase (Figure 4). We discuss these experiments in more detail below.

Finally, we have reworded the introduction to better introduce alternative hypotheses and mechanisms for paclitaxel's activity.

The authors claim that 'Previously, the anti-cancer activity of paclitaxel was thought to rely mostly on the activation of the mitotic checkpoint through disruption of microtubule dynamics, ultimately resulting in apoptosis.'

The authors may have overlooked much of the existing literature on the topic, including many recent manuscripts from Xiang-Xi Xu's and another lab.

We would like to note that the paper from Xiang-Xi Xu's lab (Smith *et al*, 2021) was cited in our original manuscript (reference 14 in both the original and revised manuscripts). We have now also included additional review articles from the Xiang-Xi Xu lab (PMID:36368286 ²⁰ and PMID: 35048083 ²¹). Furthermore, we have clarified the wording in both the introduction and discussion to better reflect the current understanding of paclitaxel's mechanism and alternative hypotheses.

The data, e.g. in Figure 1, does not hold up to the first alternative hypothesis, e.g. that paclitaxel stabilizes microtubules and that excessive mechanical bundling of microtubules induces major changes to cell shape and mechanical stress on the nucleus. Even the simplest controls for this effect (the application of an alternative MT stabilizing drug or the overexpression of an MT stabilizer, e.g., tau).

We thank the reviewer for suggesting this control experiment using the microtubule stabiliser Tau. We have now included these experiments in the revised version of the manuscript (Figure 4). The overexpression of GFP-Tau supports our hypothesis that cytoskeletal reorganisation in paclitaxel exerts mechanical stress on the nucleus during interphase, resulting in nuclear deformation and aberrations to the nuclear lamina. In particular, GFP-Tau overexpression resulted in large microtubule bundles throughout the cytoplasm (Figure 4A). Notably, in cells where these bundles extensively contacted the nucleus, we observed a significant decrease in nuclear solidity (Figure 4B) accompanied by changes in nuclear lamina organisation, including a patchy lamina phenotype, similar to that induced by paclitaxel (Figure 4C).

The focus on nuclear lamina seems somewhat arbitrary and adjacent to previously published work by other groups.

What would happen if the authors stained for focal adhesion markers? There would probably be a major change in number and distribution. Would the authors conclude that paclitaxel exerts a specific effect on focal adhesions? Or would the conclusion be that microtubule stabilization and the following mechanical disruption induce pleiotropic effects in cells? Which effects are significant for paclitaxel function on cancer cells?

We thank the reviewer for raising important points regarding the specificity of paclitaxel's effects. We agree that microtubule stabilisation can induce myriad cellular changes, including alterations to focal adhesions and other cytoskeletal components. Our focus on Lamin A/C and nuclear morphology is grounded both in the established clinical relevance of nuclear mechanics in cancer and builds on mechanistic work from other groups.

Lamin A/C expression is commonly altered in cancer, and nuclear morphology is frequently used in

cancer diagnosis³⁵. Lamin A/C also plays a crucial role in regulating nuclear mechanics³² and, importantly, determines cell sensitivity to paclitaxel¹⁴. However, the mechanism by which Lamin A/C determines sensitivity of cancer cells to paclitaxel is unclear.

Our data are consistent with Lamin A/C being a determinant of paclitaxel survival sensitivity. We also provide evidence that paclitaxel itself reduces Lamin A/C protein levels and disrupts its organisation at the nuclear envelope. We directly link these effects to microtubule bundling around the nucleus and degradation of force-sensing LINC component SUN2, highlighting the importance of nuclear architecture and mechanics to overall cellular function. Furthermore, we show that recovery from paclitaxel treatment depends on Lamin A/C expression levels. This has clinical relevance, as unlike cancer cells, healthy tissue with non-aberrant lamina would be able to selectively recover from paclitaxel treatment.

Minor comments:

While I understand the difficulty of the experiments and the effort the authors have put into producing FIB-SEM tomograms, I am not sure they are helping their study or adding anything beyond the light microscopy images. Some of the images may even be in the way, such as supplementary Figure 6, which lacks in quality, controls, and interpretation. Do I see a lot of mitochondria in that slice?

We agree with the reviewer that Supplementary Figure 6 does not add significant value to the manuscript and thank the reviewer for pointing this out. We have removed it from the manuscript accordingly.

I may have overlooked it, but has the number of cells from which lamellae have been produced been stated?

We thank the reviewer for pointing out the missing information. For our cryo-ET experiments, we collected data from 9 lamellae from paclitaxel-treated cells and 6 lamellae from control cells, with each lamella derived from a single cell. This information has now been added to the figure legend (Figure 2F).

Significance

The significance of studying the effect of paclitaxel, the most successful chemotherapy drug, should be broad and of interest to basic researchers and clinicians.

As outlined above, I believe that major concerns about the design and interpretation of the study hamper its significance and advancements.

We appreciate the reviewer's concerns and have performed major revisions to strengthen the significance of our study. Specifically, we conducted two key sets of experiments to validate our original conclusions: serum starvation to control for the effects of cell division, and overexpression of the microtubule stabiliser Tau to demonstrate that paclitaxel can affect the nucleus via its microtubule bundling activity in interphase.

By elucidating the mechanistic link between microtubule stabilisation and nuclear-cytoskeletal coupling, our findings contribute to our understanding of paclitaxel's multifaceted actions in cancer cells.

My areas of expertise could be broadly defined as Cell Biology, Cytoskeleton, Microtubules, and Structural Biology.

Reviewer #3

Evidence, reproducibility and clarity

The manuscript presents interesting new ideas for the mechanism of an old drug, taxol, which has been studied for the last 40 years.

We thank the reviewer for the positive feedback.

Although similar ideas are published, which may be suitable to be cited?

- Paclitaxel resistance related to nuclear envelope structural sturdiness. Smith ER, Wang JQ, Yang DH, Xu XX. *Drug Resist Updat.* 2022 Dec;65:100881. doi: 10.1016/j.drup.2022.100881. Epub 2022 Oct 15. PMID: 36368286 Review.
- Breaking malignant nuclei as a non-mitotic mechanism of taxol/paclitaxel. Smith ER, Xu XX. *J Cancer Biol.* 2021;2(4):86-93. doi: 10.46439/cancerbiology.2.031. PMID: 35048083 Free PMC article.

We thank the reviewer for bringing to our attention these important review articles. In our initial manuscript, we only cited the original paper (¹⁴, also reference 14 in the original manuscript). We have now included citations to the suggested publications (^{20,21}).

We would also like to emphasise how our manuscript distinguishes itself from the work of Smith *et al.* ^{14,20,21}:

- § **Cell type focus:** In their study¹⁴, Smith *et al.* examined the effect of paclitaxel on malignant ovarian cancer cells and proposed that paclitaxel's effects on the nucleus are limited to cancer cells. However, our data extends these findings by demonstrating paclitaxel's effects in both cancerous and non-cancerous backgrounds.
- § **Cytoskeletal reorganisation:** Smith *et al.* show reorganisation of microtubules in paclitaxel treated cells¹⁴. Our data show re-organisation of other cytoskeletal components, including F-actin and vimentin.
- § **Multimicronucleation:** Smith *et al.* propose that paclitaxel-induced multimicronucleation occurs independently of cell division¹⁴. Although we observe progressive nuclear abnormalities during interphase over the course of paclitaxel treatment, our data do not support this conclusion; we find that multimicronucleation occurs only following mitosis.
- § **Direct link between microtubule bundling and nuclear aberrations:** We show that nuclear aberrations caused by paclitaxel during interphase (distinct from multimicronucleation) are directly linked to microtubule bundling around the nucleus, suggesting they result from mechanical disruption and altered force propagation.
- § **Lamin A/C regulation:** Consistent with Smith *et al.*¹⁴, we show that Lamin A/C depletion leads to increased sensitivity to paclitaxel treatment. However, we further demonstrate that paclitaxel itself leads to reduced levels of Lamin A/C and that this effect occurs independently of mitosis and is mediated via force-sensing LINC component SUN2. Upon SUN2 knockdown, Lamin A/C levels are no longer affected by paclitaxel treatment.
- § **Recovery:** Finally, our work reveals that cells expressing low levels of Lamin A/C recover less efficiently after paclitaxel removal. This might help explain how cancer cells could be more susceptible to paclitaxel.

Only one cell line was used in all the experiments? "Human telomerase reverse transcriptase (hTERT) immortalised human fibroblasts" ? The cells used are not very relevant to cancer cells (carcinomas) that are treated with paclitaxel. It is not clear if the observations and conclusions will be able to be generalized to cancer cells.

We thank the reviewer for this comment. Our initial study aimed to understand the effects of paclitaxel on nuclear architecture in non-aberrant backgrounds. To show that the observed effects of paclitaxel are also applicable to cancer cells, we have now repeated our main experiments using MDA-MB-231 human breast cancer cells (**Supplementary Figure 1B; Supplementary Figure 3P-T**). Similar to our findings in human fibroblasts, paclitaxel treatment of MDA-MB-231 led to cytoskeletal reorganisation (**Supplementary Figure 1B**), decrease in nuclear solidity (**Supplementary Figure 3P**), aberrant (patchy) localisation of Lamin A/C (**Supplementary Figure 3Q**), and reduction in Lamin A/C and SUN2 levels (**Supplementary Figure 3R-T**).

"Fig. 1. (B) STORM imaging of α -tubulin immunofluorescence in cells fixed after 16 h incubation in control media or 5 nM paclitaxel. Lower panels show α -tubulin clusters generated with HDBSCAN analysis. Scale bars = 10 μ m."

It needs explanation of what is meaning of the different color lines in the lower panels, just different filaments?

We have added further detail to the figure legend for clarification: "*Lower panels show α -tubulin clusters generated with HDBSCAN analysis. Different colours distinguish individual α -tubulin clusters, representing individual microtubule filaments or filament bundles.*"

Generally, the figures need additional description to be clear.

We have added further clarification and detail to our figure legends.

"Figure 3 - Paclitaxel results in aberrations to the nuclear lamina."

The sentence seems not to be well constructed. "Paclitaxel treatment causes ..."?

We changed this sentence to: "*Figure 3 - Paclitaxel treatment results in aberrant organisation of the nuclear lamina and decreased Lamin A/C levels via SUN2.*"

Lamin A and C levels are different in different images (Fig. 3B, H): some Lamin A is higher, and sometime Lamin C is higher? This may possibly due to culture condition or subtle difference in sample handling?.

We thank the reviewer for pointing this out and we agree that the ratio of Lamin A to Lamin C can vary with culture conditions. To confirm that paclitaxel treatment reduces total Lamin A/C levels regardless of this ratio, we repeated the Western blot analysis in three additional biological replicates using cells in which Lamin C levels exceeded Lamin A levels. These experiments confirmed a comparable decrease in total Lamin A/C levels. Figure 3B and 3C have been updated accordingly.

Also, the effect on Lamin A/C and SUN2 levels are not significant of robust.

Decreased Lamin A/C and SUN2 levels following paclitaxel treatment were consistently seen across three or more biological repeats (**Figure 3B-C**), and this could be replicated in a different cell type (MDA-MB-231) (**Supplementary Figure 3R-T**). Furthermore, Western blotting results are consistent with the patchy Lamin A/C distribution observed using confocal and STORM following paclitaxel treatment (Figure 3A; **Supplementary Figure 3A**), where Lamin A/C appears to be absent from discrete areas of the lamina.

Any mechanisms are speculated for the reason for the reduction?

We have now included additional data which aims to shed light on the mechanism behind the decrease in Lamin A/C and SUN2 levels following paclitaxel treatment. We found that SUN2 is selectively degraded during paclitaxel treatment. Immunoprecipitation of SUN2 followed by Western blotting against Polyubiquitin C showed increased SUN2 ubiquitination in paclitaxel (**Figure 3M and N**). Furthermore, in our original manuscript we showed that Lamina A/C levels remained unaltered during paclitaxel treatment in cells where SUN2 had been knocked down. We propose that changes in microtubule organisation affect force propagation to Lamin A/C specifically via SUN2 and that this leads to Lamina A/C removal and depletion. Future work will be needed to fully understand this mechanism.

In addition to the findings described above, we report no significant changes in mRNA levels for *LMNA* or *SUN2* in paclitaxel (**Supplementary Figure 3B and O**). Phos-tag gels followed by Western blotting analysis for Lamin A/C also did not detect changes to the overall phosphorylation status of Lamin A/C due to paclitaxel treatment. This is in agreement with our initial data showing no changes to Lamin A/C Ser 404 phosphorylation levels (**Supplementary Figure 3E and F**). Finally, Lamin A/C immunoprecipitation experiments followed by Western blotting for Polyubiquitin C and acetyl-lysine showed no significant changes in the ubiquitination and acetylation state of Lamin

A/C in paclitaxel-treated cells (Supplementary Figure 3G-I).

Also, the about 50% reduction in protein level is difficult to be convincing as an explanation of nuclear disruption.

The nuclear lamina and LINC complex proteins play a critical role in regulating nuclear integrity, stiffness and mechanical responsiveness to external forces^{28,31-33,54,75}, as well as in maintaining the nuclear intermembrane distance^{69,74}. In particular, SUN-domain proteins physically bridge the nuclear lamina to the cytoskeleton through interactions with Nesprins, thereby preserving the perinuclear space distance^{30,69,74}. Mutations in Lamins have been shown to disrupt chromatin organization, alter gene expression, and compromise nuclear structural integrity, and experiments with LMNA knockout cells reveal that nuclear mechanical fragility is closely coupled to nuclear deformation⁴⁷. Furthermore, nuclear-cytoskeletal coupling is essential during processes such as cell migration, where cells undergo stretching and compression of the nucleus; weakening or loss of the lamina in such cases compromises cell movement^{47,73}. In our work, we show that alterations to nuclear Lamin A/C and SUN2 by paclitaxel treatment coincide with nuclear deformations (Figure 2A-D, F, G; Figure 3A-D, F, G; Supplementary Figure 3A, P-T) and that these deformations are reversible following paclitaxel removal (Supplementary Figure 4B-D). Our experiments also demonstrate that Lamin A/C expression levels significantly influence cell growth, cell viability, and cell recovery in paclitaxel (Figure 5).

Therefore, drawing on current literature and our results, we propose that, during interphase, paclitaxel induces severe nuclear aberrations through the combined effects of: i) increased cytoskeletal forces on the NE caused by microtubule bundling; ii) loss of ~50% Lamin A/C and SUN2; iii) reorganisation of nucleo-cytoskeletal components.

Significance

The manuscript presents interesting new ideas for the mechanism of an old drug, taxol, which has been studied for the last 40 years.

The data may be improved to provide stronger support.

Additional cell lines (of cancer or epithelial origin) may be repeated to confirm the generality of the observation and conclusions.?

We thank the reviewer for the feedback and valuable suggestions. In response, we have included experiments using human breast cancer cell line MDA-MB-231 to further corroborate our findings and interpretations. We believe these additions have improved the clarity, robustness and impact of our manuscript, and we are grateful for the reviewer's contributions to its improvement.

Original submission

First decision letter

MS ID#: jcs.264494

MS TITLE: Paclitaxel compromises nuclear integrity in interphase through SUN2-mediated cytoskeletal coupling

AUTHORS: Thomas Hale; Victoria Louise Hale; Piotr Kolata; Alia dos Santos; Matteo Allegretti
ARTICLE TYPE: Review Commons Transfer

Dear Dr Allegretti,

Many thanks for transferring your manuscript to Journal of Cell Science from Review Commons. I have now had the chance to review your documents. I am enthusiastic about your manuscript and have been able to assess it without further input from the referees. I would therefore like to invite you to prepare a final version of the manuscript that incorporates our best practices for data transparency including showing individual points in bar graphs and violin plots, which you use extensively in the manuscript.

First revision

Author response to reviewers' comments

1. General Statements [optional]

This section is optional. Insert here any general statements you wish to make about the goal of the study or about the reviews.

In this study, we investigated the effects of paclitaxel on both healthy and cancerous cells, focusing on alterations in nuclear architecture. Our novel findings show that:

- Paclitaxel-induced microtubule reorganisation during interphase alters the perinuclear distribution of actin and vimentin. The formation of extensive microtubule bundles, in paclitaxel or following GFP-Tau overexpression, coincides with nuclear shape deformation, loss of regulation of nuclear envelope spacing, and alteration of the nuclear lamina.
- Paclitaxel treatment reduces Lamin A/C protein levels via a SUN2-dependent mechanism. SUN2, which links the lamina to the cytoskeleton, undergoes ubiquitination and consequent degradation following paclitaxel exposure.
- Lamin A/C expression, frequently dysregulated in cancer cells, is a key determinant of cellular sensitivity to, and recovery from, paclitaxel treatment.

Collectively, our data support a model in which paclitaxel disrupts nuclear architecture through two mechanisms: (i) aberrant nuclear-cytoskeletal coupling during interphase, and (ii) multimicronucleation following defective mitotic exit. This represents an additional mode of action for paclitaxel beyond its well-established mechanism of mitotic arrest.

We thank the reviewers for their time and constructive feedback. We have carefully considered all comments and have carried out a full revision. The updated manuscript now includes additional data showing:

- § Overexpression of microtubule-associated protein Tau causes similar nuclear aberration phenotypes to paclitaxel. This supports our hypothesis that increased microtubule bundling directly leads to nuclear disruption in paclitaxel during interphase.
- § Paclitaxel's effects on nuclear shape and Lamin A/C and SUN2 expression levels occur independently of cell division.
- § Reduced levels of Lamin A/C and SUN2 upon paclitaxel treatment occur at the protein level via ubiquitination of SUN2.
- § The effects of paclitaxel on the nucleus are conserved in breast cancer cells.

We have also edited our text and added further detail to clarify points raised by the reviewers. We believe that our revised manuscript is overall more complete, solid and compelling thanks to the reviewers' comments.

This section is mandatory. Please insert a point-by-point reply describing the revisions that were

already carried out and included in the transferred manuscript.

Reviewer #1

Evidence, reproducibility and clarity

This description of the down-regulation of the expression of lamin A/C upon treatment with paclitaxel and its sensitivity to SUN2 is quite interesting but still somehow preliminary. It is unclear whether this effect involves the regulation of gene expression, or of the stability of the proteins. How SUN2 mediates this effect is still unknown.

We thank the reviewer for this valuable comment. To elucidate the mechanism behind the decrease in Lamin A/C and SUN2 levels, we have now performed several additional experiments. First, we performed RT-qPCR to quantify mRNA levels of these genes, relative to the housekeeping gene *GAPDH* (**Supplementary Figure 3B and O**). The levels of *SUN2* and *LMNA* mRNA remained the same between control and paclitaxel-treated cells, indicating that this effect instead occurs at the protein level. We have also tested post-translational modifications as a potential regulatory mechanism for Lamin A/C and SUN2. In addition to the phosphorylation of Ser404 which we had already tested (Supplementary Figure 3C), we have now included additional Phos-tag gel and Western blotting data showing that the overall phosphorylation status of Lamin A/C is not affected by paclitaxel (**Supplementary Figure 3E and F**). We also pulled-down Lamin A/C from cell lysates and then Western blotted for polyubiquitin and acetyl-lysine, which showed that the ubiquitination and acetylation states of Lamin A/C are also not affected by paclitaxel (**Supplementary Figure 3G-I**). However, Western blots for polyubiquitin of SUN2 pulled down from cell lysates showed that paclitaxel treatment results in significant SUN2 ubiquitination (**Figure 3M and N**). Therefore, we propose that the downregulation of SUN2 following paclitaxel treatment occurs by ubiquitin-mediated proteolysis.

The roles of free tubulins and polymerized microtubules, and thus the potential role of paclitaxel, need to be uncovered.

We addressed this important point by using an alternative method to stabilise/bundle microtubules in interphase, namely by overexpressing GFP-Tau, as suggested by reviewer 2. Following GFP-Tau overexpression, large microtubule bundles were observed throughout the cytoplasm (**Figure 4A**), and this resulted in a significant decrease in nuclear solidity (**Figure 4B**). Furthermore, in cells where microtubule bundles extensively contacted the nucleus, the nuclear lamina became unevenly distributed and appeared patchy (**Figure 4C**). This supports our hypothesis that the aberrations to nuclear shape and Lamin A/C localisation in paclitaxel-treated cells are due to the presence of microtubules bundles surrounding the nucleus.

The doses of paclitaxel at which occur the effects described in the paper are not fully consistent with all the conclusions. Most experiments have been done at 5 nM. However, at this dose the effect of lamin A/C over or down expression on the growth (differences in the slopes of the curves in **Figure 4A**) are not fully convincing and not fully consistent with the clear effect on viability as well (in addition, duration of treatments before assessing viability are not specified). At 1 nM, cell growth is reduced and the rescuing effect of lamin over-expression is much more clear (**Fig 4A**), and the nucleus deformation clear (**Fig 2A**) but this dose has no effect on lamin A/C expression (**Fig 3C**), which questions how lamins impact nucleus shape and cell survival. Cytoskeleton reorganisation in these conditions is not described although it could clarify the respective role of force production (suggested in **figure 1**) and nuclei resistance (shown in **figure 2**) in paclitaxel sensitivity.

We thank the reviewer for raising this important point. We have addressed this by conducting additional repeats for the cell confluency measurements to increase the statistical power of our experiments (**Figure 5A**). Our data now show that GFP-lamin A/C had a statistically significant effect on rescuing cell growth at both 1 nM and 5 nM paclitaxel, while Lamin A/C knockdown exacerbated the inhibition of cell growth at 5 nM paclitaxel but not 1 nM paclitaxel (**Figure 5A**). In addition, we note that the duration of paclitaxel treatment before assessing viability was specified in the figure legend: “Bar graph comparing cell viability between wild-type (red), GFP-Lamin A/C overexpression (green), and Lamin A/C knockdown (blue) cells following 20 h incubation in 0, 1, 5,

or 10 nM paclitaxel.” We also repeated cell viability analysis after 48 h incubation in paclitaxel instead of 20 h to allow for a longer time for differences to take effect (Figure 5B).

We also added figures showing the cytoskeletal reorganisation at both 1 and 10 nM in addition to 0 and 5 nM (Supplementary Figure 1A) showing that microtubule bundling and condensation of actin into puncta correlated with increased paclitaxel concentration. Vimentin colocalised well with microtubules at all concentrations.

We have also included in our results section further clarification for the use of 5nM paclitaxel in this study. The new section reads as follows: “Experiments were performed at 5 nM paclitaxel (with additional experiments to determine dose relationships at 1 and 10 nM) because this aligns with previous studies^{7,14,24}. Furthermore, previous analysis of patient plasma reveals that typical concentrations are within the low nanomolar range⁸, and concentrations of 5-10 nM are required in cell culture to reach the same intracellular concentrations observed in vivo in patient tumours⁹. This aligns with in vitro cytotoxic studies of paclitaxel in eight human tumour cell lines which show that paclitaxel’s IC50 ranges between 2.5 and 7.5 nM⁴¹.”

Finally, although the absence of role of mitotic arrest is clear from the data, the defective reorganisation of the nucleus after mitosis still suggest that the effect of paclitaxel is not independent of mitosis.

We thank the reviewer for pointing out the need for clarification in the wording of our manuscript. We have reworded the title and relevant sections of our abstract, introduction, and discussion to make it clearer that the effects of paclitaxel on the nucleus are due to a combination of aberrant nuclear cytoskeletal coupling during interphase and multimicronucleation following mitotic slippage. We have also added additional data in support of the effect of paclitaxel on nuclear architecture during interphase. For this, we used serum-starved cells (which divide only very slowly such that the majority of cells do not pass through mitosis during the 16 h incubation in paclitaxel [Supplementary Figure 2D]). Our new data confirmed that paclitaxel’s effects on nuclear solidity, and Lamin A/C and SUN2 proteins levels can occur independently of cell division (Figure 2C; Figure 3H-J). Finally, when we overexpressed GFP-Tau (as discussed above) we observed similar aberrations to nuclear solidity and Lamin A/C localisation. This indicates that these effects occur due to microtubule bundling in interphase, especially as in our study GFP-Tau did not lead to multimicronucleation or appear to affect mitosis (Figure 4).

Below are the main changes to the text regarding the interphase effect of paclitaxel:

- Title: “Paclitaxel compromises nuclear integrity in interphase through SUN2-mediated cytoskeletal coupling”
- Abstract: “Our findings support a model in which paclitaxel acts through both defective mitosis and interphase nuclear-cytoskeletal disruption, providing additional mechanistic insights into a widely used anticancer drug.”
- Introduction: “Here we propose that cancer cells have increased vulnerability to paclitaxel both during interphase and following aberrant mitosis due to pre-existing defects in their NE and nuclear lamina.”
- Discussion: “Overall, our work builds on previous studies investigating loss of nuclear integrity as an anti-cancer mechanism of paclitaxel separate from mitotic arrest^{14,20,21}. We propose that cancer cells show increased sensitivity to nuclear deformation induced by aberrant nuclear-cytoskeletal coupling and multimicronucleation following mitotic slippage. Therefore, we conclude that paclitaxel functions in interphase as well as mitosis, elucidating how slowly growing tumours are targeted.”

minor: a more thorough introduction of known data about dose response of cells in culture and in vivo would help understanding the range of concentrations used in this study.

As mentioned above, we have now included additional information in our Results section to clarify our paclitaxel dose range:

“Experiments were performed at 5 nM paclitaxel (with additional experiments to determine dose

relationships at 1 and 10 nM) because this aligns with previous studies^{7,14,24}. Furthermore, previous analysis of patient plasma reveals that typical concentrations are within the low nanomolar range⁸, and concentrations of 5-10 nM are required in cell culture to reach the same intracellular concentrations observed in vivo in patient tumours⁹. This aligns with in vitro cytotoxic studies of paclitaxel in eight human tumour cell lines which show that paclitaxel's IC50 ranges between 2.5 and 7.5 nM⁴¹."

Significance

In this manuscript, Hale and colleagues describe the effect of paclitaxel on nucleus deformation and cell survival. They showed that 5nM of paclitaxel induces nucleus fragmentation, cytoskeleton reorganisation, reduced expression of LaminA/C and SUN2, and reduced cell growth and viability. They also showed that these effects could be at least partly compensated by the over-expression of lamin A/C. As fairly acknowledged by the authors, the induction of nuclear deformation in paclitaxel-treated cells, and the increased sensitivity to paclitaxel of cells expressing low level of lamin A/C are not novel (reference #14). Here the authors provided more details on the cytoskeleton changes and nuclear membrane deformation upon paclitaxel treatment. The effect of lamin A/C over and down expression on cell growth and survival are not fully convincing, as further discussed below. The most novel part is the observation that paclitaxel can induce the down-regulation of the expression of lamin A/C and that this effect is mediated by SUN2.

We appreciate the reviewer's summary and thank them for their time. We believe our comprehensive revisions have addressed all comments, strengthening the manuscript and making it more robust and compelling.

Reviewer #2

Evidence, reproducibility and clarity

This study investigates the effects of the chemotherapeutic drug paclitaxel on nuclear-cytoskeletal coupling during interphase, claiming a novel mechanism for its anti-cancer activity. The study uses hTERT-immortalized human fibroblasts. After paclitaxel exposure, a suite of state-of-the-art imaging modalities visualizes changes in the cytoskeleton and nuclear architecture. These include STORM imaging and a large number of FIB-SEM tomograms.

We thank the reviewer for the summary and for highlighting our efforts in using the latest imaging technical advances.

Major comments:

The authors make a major claim that in addition to the somewhat well-described mechanism of paclitaxel on mitosis, they have discovered 'an alternative, poorly characterised mechanism in interphase'.

However, none of the data proves that the effects shown are independent of mitosis. To the contrary, measurements are presented 48 hours after paclitaxel treatment starts, after which it can be assumed that 100% of cells have completed at least one mitotic event. The appearance of micronuclei evidences this, as discussed by the authors shortly. It looks like most of the results shown are based on botched mitosis or, more specifically, errors on nuclear assembly upon exit from mitosis rather than a specific effect of paclitaxel on interphase. The readouts the authors show just happen to be measurements while the cells are in interphase. Alternative hypotheses are missing throughout the manuscript, and so are critical controls and interpretations.

We thank the reviewer for highlighting the lack of clarity in our wording. We have revised the title, abstract and relevant sections of the introduction and discussion to clarify our message that the

effects of paclitaxel on the nucleus arise from a combination of aberrant nuclear-cytoskeletal coupling during interphase and multimicronucleation following exit from defective mitosis. We have also included additional data where we used slow-dividing, serum-starved cells (under these conditions, the majority of cells do not undergo mitosis during the 16 h incubation in paclitaxel [Supplementary Figure 2D]). Our new data show that even in these cells there is a clear effect of paclitaxel on nuclear solidity, and Lamin A/C and SUN2 protein levels, further supporting our hypothesis that these phenotypes can occur independently of cell division (Figure 2C; Figure 3H-J).

Furthermore, we performed additional experiments where we used overexpression of GFP-Tau as an alternative method of stabilising microtubules in interphase and observed similar aberrations to nuclear solidity and Lamin A/C localisation. As GFP-Tau overexpression did not lead to micronucleation or appear to affect mitosis, these data support the hypothesis that nuclear aberrations occur due to microtubule bundling in interphase (Figure 4). We discuss these experiments in more detail below.

Finally, we have reworded the introduction to better introduce alternative hypotheses and mechanisms for paclitaxel's activity.

The authors claim that 'Previously, the anti-cancer activity of paclitaxel was thought to rely mostly on the activation of the mitotic checkpoint through disruption of microtubule dynamics, ultimately resulting in apoptosis.'

The authors may have overlooked much of the existing literature on the topic, including many recent manuscripts from Xiang-Xi Xu's and another lab.

We would like to note that the paper from Xiang-Xi Xu's lab (Smith *et al*, 2021) was cited in our original manuscript (reference 14 in both the original and revised manuscripts). We have now also included additional review articles from the Xiang-Xi Xu lab (PMID:36368286 ²⁰ and PMID: 35048083 ²¹). Furthermore, we have clarified the wording in both the introduction and discussion to better reflect the current understanding of paclitaxel's mechanism and alternative hypotheses.

The data, e.g. in Figure 1, does not hold up to the first alternative hypothesis, e.g. that paclitaxel stabilizes microtubules and that excessive mechanical bundling of microtubules induces major changes to cell shape and mechanical stress on the nucleus. Even the simplest controls for this effect (the application of an alternative MT stabilizing drug or the overexpression of an MT stabilizer, e.g., tau).

We thank the reviewer for suggesting this control experiment using the microtubule stabiliser Tau. We have now included these experiments in the revised version of the manuscript (Figure 4). The overexpression of GFP-Tau supports our hypothesis that cytoskeletal reorganisation in paclitaxel exerts mechanical stress on the nucleus during interphase, resulting in nuclear deformation and aberrations to the nuclear lamina. In particular, GFP-Tau overexpression resulted in large microtubule bundles throughout the cytoplasm (Figure 4A). Notably, in cells where these bundles extensively contacted the nucleus, we observed a significant decrease in nuclear solidity (Figure 4B) accompanied by changes in nuclear lamina organisation, including a patchy lamina phenotype, similar to that induced by paclitaxel (Figure 4C).

The focus on nuclear lamina seems somewhat arbitrary and adjacent to previously published work by other groups.

What would happen if the authors stained for focal adhesion markers? There would probably be a major change in number and distribution. Would the authors conclude that paclitaxel exerts a specific effect on focal adhesions? Or would the conclusion be that microtubule stabilization and the following mechanical disruption induce pleiotropic effects in cells? Which effects are significant for paclitaxel function on cancer cells?

We thank the reviewer for raising important points regarding the specificity of paclitaxel's effects.

We agree that microtubule stabilisation can induce myriad cellular changes, including alterations to focal adhesions and other cytoskeletal components. Our focus on Lamin A/C and nuclear morphology is grounded both in the established clinical relevance of nuclear mechanics in cancer and builds on mechanistic work from other groups.

Lamin A/C expression is commonly altered in cancer, and nuclear morphology is frequently used in cancer diagnosis³⁵. Lamin A/C also plays a crucial role in regulating nuclear mechanics³² and, importantly, determines cell sensitivity to paclitaxel¹⁴. However, the mechanism by which Lamin A/C determines sensitivity of cancer cells to paclitaxel is unclear.

Our data are consistent with Lamin A/C being a determinant of paclitaxel survival sensitivity. We also provide evidence that paclitaxel itself reduces Lamin A/C protein levels and disrupts its organisation at the nuclear envelope. We directly link these effects to microtubule bundling around the nucleus and degradation of force-sensing LINC component SUN2, highlighting the importance of nuclear architecture and mechanics to overall cellular function. Furthermore, we show that recovery from paclitaxel treatment depends on Lamin A/C expression levels. This has clinical relevance, as unlike cancer cells, healthy tissue with non-aberrant lamina would be able to selectively recover from paclitaxel treatment.

Minor comments:

While I understand the difficulty of the experiments and the effort the authors have put into producing FIB-SEM tomograms, I am not sure they are helping their study or adding anything beyond the light microscopy images. Some of the images may even be in the way, such as supplementary Figure 6, which lacks in quality, controls, and interpretation. Do I see a lot of mitochondria in that slice?

We agree with the reviewer that Supplementary Figure 6 does not add significant value to the manuscript and thank the reviewer for pointing this out. We have removed it from the manuscript accordingly.

I may have overlooked it, but has the number of cells from which lamellae have been produced been stated?

We thank the reviewer for pointing out the missing information. For our cryo-ET experiments, we collected data from 9 lamellae from paclitaxel-treated cells and 6 lamellae from control cells, with each lamella derived from a single cell. This information has now been added to the figure legend (Figure 2F).

Significance

The significance of studying the effect of paclitaxel, the most successful chemotherapy drug, should be broad and of interest to basic researchers and clinicians.

As outlined above, I believe that major concerns about the design and interpretation of the study hamper its significance and advancements.

We appreciate the reviewer's concerns and have performed major revisions to strengthen the significance of our study. Specifically, we conducted two key sets of experiments to validate our original conclusions: serum starvation to control for the effects of cell division, and overexpression of the microtubule stabiliser Tau to demonstrate that paclitaxel can affect the nucleus via its microtubule bundling activity in interphase.

By elucidating the mechanistic link between microtubule stabilisation and nuclear-cytoskeletal coupling, our findings contribute to our understanding of paclitaxel's multifaceted actions in cancer cells.

My areas of expertise could be broadly defined as Cell Biology, Cytoskeleton, Microtubules, and

Structural Biology.

Reviewer #3

Evidence, reproducibility and clarity

The manuscript presents interesting new ideas for the mechanism of an old drug, taxol, which has been studied for the last 40 years.

We thank the reviewer for the positive feedback.

Although similar ideas are published, which may be suitable to be cited?

- Paclitaxel resistance related to nuclear envelope structural sturdiness. Smith ER, Wang JQ, Yang DH, Xu XX. Drug Resist Updat. 2022 Dec;65:100881. doi: 10.1016/j.drug.2022.100881. Epub 2022 Oct 15. PMID: 36368286 Review.
- Breaking malignant nuclei as a non-mitotic mechanism of taxol/paclitaxel. Smith ER, Xu XX. J Cancer Biol. 2021;2(4):86-93. doi: 10.46439/cancerbiology.2.031. PMID: 35048083 Free PMC article.

We thank the reviewer for bringing to our attention these important review articles. In our initial manuscript, we only cited the original paper (¹⁴, also reference 14 in the original manuscript). We have now included citations to the suggested publications (^{20,21}).

We would also like to emphasise how our manuscript distinguishes itself from the work of Smith *et al.* ^{14,20,21}:

- § **Cell type focus:** In their study¹⁴, Smith *et al.* examined the effect of paclitaxel on malignant ovarian cancer cells and proposed that paclitaxel's effects on the nucleus are limited to cancer cells. However, our data extends these findings by demonstrating paclitaxel's effects in both cancerous and non-cancerous backgrounds.
- § **Cytoskeletal reorganisation:** Smith *et al.* show reorganisation of microtubules in paclitaxel treated cells¹⁴. Our data show re-organisation of other cytoskeletal components, including F-actin and vimentin.
- § **Multimicronucleation:** Smith *et al.* propose that paclitaxel-induced multimicronucleation occurs independently of cell division¹⁴. Although we observe progressive nuclear abnormalities during interphase over the course of paclitaxel treatment, our data do not support this conclusion; we find that multimicronucleation occurs only following mitosis.
- § **Direct link between microtubule bundling and nuclear aberrations:** We show that nuclear aberrations caused by paclitaxel during interphase (distinct from multimicronucleation) are directly linked to microtubule bundling around the nucleus, suggesting they result from mechanical disruption and altered force propagation.
- § **Lamin A/C regulation:** Consistent with Smith *et al.* ¹⁴, we show that Lamin A/C depletion leads to increased sensitivity to paclitaxel treatment. However, we further demonstrate that paclitaxel itself leads to reduced levels of Lamin A/C and that this effect occurs independently of mitosis and is mediated via force-sensing LINC component SUN2. Upon SUN2 knockdown, Lamin A/C levels are no longer affected by paclitaxel treatment.
- § **Recovery:** Finally, our work reveals that cells expressing low levels of Lamin A/C recover less efficiently after paclitaxel removal. This might help explain how cancer cells could be more susceptible to paclitaxel.

Only one cell line was used in all the experiments? "Human telomerase reverse transcriptase (hTERT) immortalised human fibroblasts" ? The cells used are not very relevant to cancer cells (carcinomas) that are treated with paclitaxel. It is not clear if the observations and conclusions will be able to be generalized to cancer cells.

We thank the reviewer for this comment. Our initial study aimed to understand the effects of paclitaxel on nuclear architecture in non-aberrant backgrounds. To show that the observed effects of paclitaxel are also applicable to cancer cells, we have now repeated our main experiments using MDA-MB-231 human breast cancer cells (**Supplementary Figure 1B**; **Supplementary Figure 3P-T**). Similar to our findings in human fibroblasts, paclitaxel treatment of MDA-MB-231 led to cytoskeletal reorganisation (**Supplementary Figure 1B**), decrease in nuclear solidity (**Supplementary Figure 3P**), aberrant (patchy) localisation of Lamin A/C (**Supplementary Figure 3Q**), and reduction in Lamin A/C and SUN2 levels (**Supplementary Figure 3R-T**).

"Fig. 1. (B) STORM imaging of α -tubulin immunofluorescence in cells fixed after 16 h incubation in control media or 5 nM paclitaxel. Lower panels show α -tubulin clusters generated with HDBSCAN analysis. Scale bars = 10 μ m."

It needs explanation of what is meaning of the different color lines in the lower panels, just different filaments?

We have added further detail to the figure legend for clarification: "*Lower panels show α -tubulin clusters generated with HDBSCAN analysis. Different colours distinguish individual α -tubulin clusters, representing individual microtubule filaments or filament bundles.*"

Generally, the figures need additional description to be clear.

We have added further clarification and detail to our figure legends.

"Figure 3 - Paclitaxel results in aberrations to the nuclear lamina."

The sentence seems not to be well constructed. "Paclitaxel treatment causes ..."?

We changed this sentence to: "*Figure 3 - Paclitaxel treatment results in aberrant organisation of the nuclear lamina and decreased Lamin A/C levels via SUN2.*"

Lamin A and C levels are different in different images (Fig. 3B, H): some Lamin A is higher, and sometime Lamin C is higher? This may possibly due to culture condition or subtle difference in sample handling?.

We thank the reviewer for pointing this out and we agree that the ratio of Lamin A to Lamin C can vary with culture conditions. To confirm that paclitaxel treatment reduces total Lamin A/C levels regardless of this ratio, we repeated the Western blot analysis in three additional biological replicates using cells in which Lamin C levels exceeded Lamin A levels. These experiments confirmed a comparable decrease in total Lamin A/C levels. Figure 3B and 3C have been updated accordingly.

Also, the effect on Lamin A/C and SUN2 levels are not significant of robust.

Decreased Lamin A/C and SUN2 levels following paclitaxel treatment were consistently seen across three or more biological repeats (**Figure 3B-C**), and this could be replicated in a different cell type (MDA-MB-231) (**Supplementary Figure 3R-T**). Furthermore, Western blotting results are consistent with the patchy Lamin A/C distribution observed using confocal and STORM following paclitaxel treatment (Figure 3A; **Supplementary Figure 3A**), where Lamin A/C appears to be absent from discrete areas of the lamina.

Any mechanisms are speculated for the reason for the reduction?

We have now included additional data which aims to shed light on the mechanism behind the decrease in Lamin A/C and SUN2 levels following paclitaxel treatment. We found that SUN2 is selectively degraded during paclitaxel treatment. Immunoprecipitation of SUN2 followed by Western blotting against Polyubiquitin C showed increased SUN2 ubiquitination in paclitaxel (**Figure 3M and N**). Furthermore, in our original manuscript we showed that Lamina A/C levels remained unaltered during paclitaxel treatment in cells where SUN2 had been knocked down. We propose that changes in microtubule organisation affect force propagation to Lamin A/C specifically via SUN2

and that this leads to Lamina A/C removal and depletion. Future work will be needed to fully understand this mechanism.

In addition to the findings described above, we report no significant changes in mRNA levels for *LMNA* or *SUN2* in paclitaxel (**Supplementary Figure 3B and O**). Phos-tag gels followed by Western blotting analysis for Lamin A/C also did not detect changes to the overall phosphorylation status of Lamin A/C due to paclitaxel treatment. This is in agreement with our initial data showing no changes to Lamin A/C Ser 404 phosphorylation levels (**Supplementary Figure 3E and F**). Finally, Lamin A/C immunoprecipitation experiments followed by Western blotting for Polyubiquitin C and acetyl-lysine showed no significant changes in the ubiquitination and acetylation state of Lamin A/C in paclitaxel-treated cells (**Supplementary Figure 3G-I**).

Also, the about 50% reduction in protein level is difficult to be convincing as an explanation of nuclear disruption.

The nuclear lamina and LINC complex proteins play a critical role in regulating nuclear integrity, stiffness and mechanical responsiveness to external forces^{28,31-33,54,75}, as well as in maintaining the nuclear intermembrane distance^{69,74}. In particular, SUN-domain proteins physically bridge the nuclear lamina to the cytoskeleton through interactions with Nesprins, thereby preserving the perinuclear space distance^{30,69,74}. Mutations in Lamins have been shown to disrupt chromatin organization, alter gene expression, and compromise nuclear structural integrity, and experiments with LMNA knockout cells reveal that nuclear mechanical fragility is closely coupled to nuclear deformation⁴⁷. Furthermore, nuclear-cytoskeletal coupling is essential during processes such as cell migration, where cells undergo stretching and compression of the nucleus; weakening or loss of the lamina in such cases compromises cell movement^{47,73}. In our work, we show that alterations to nuclear Lamin A/C and SUN2 by paclitaxel treatment coincide with nuclear deformations (Figure 2A-D, F, G; Figure 3A-D, F, G; Supplementary Figure 3A, P-T) and that these deformations are reversible following paclitaxel removal (Supplementary Figure 4B-D). Our experiments also demonstrate that Lamin A/C expression levels significantly influence cell growth, cell viability, and cell recovery in paclitaxel (Figure 5). Therefore, drawing on current literature and our results, we propose that, during interphase, paclitaxel induces severe nuclear aberrations through the combined effects of: i) increased cytoskeletal forces on the NE caused by microtubule bundling; ii) loss of ~50% Lamin A/C and SUN2; iii) reorganisation of nucleo-cytoskeletal components.

Significance

The manuscript presents interesting new ideas for the mechanism of an old drug, taxol, which has been studied for the last 40 years.

The data may be improved to provide stronger support.

Additional cell lines (of cancer or epithelial origin) may be repeated to confirm the generality of the observation and conclusions.?

We thank the reviewer for the feedback and valuable suggestions. In response, we have included experiments using human breast cancer cell line MDA-MB-231 to further corroborate our findings and interpretations. We believe these additions have improved the clarity, robustness and impact of our manuscript, and we are grateful for the reviewer's contributions to its improvement.

Second decision letter

MS ID#: jcs.264494R1

MS Title: Paclitaxel compromises nuclear integrity in interphase through SUN2-mediated cytoskeletal coupling

Authors: Thomas Hale; Victoria Louise Hale; Piotr Kolata; Alia dos Santos; Matteo Allegretti
Article Type: Review Commons Transfer

Dear Dr Allegretti,

I am happy to tell you that your manuscript has been accepted for publication in Journal of Cell Science, pending standard publication integrity checks.

Thank you for sending your manuscript to Journal of Cell Science through Review Commons.